# Genetic and morphological divergence in the warm-water planktonic foraminifera genus *Globigerinoides*

**Raphaël Morard**[1]*, **Angelina Füllberg**[1], **Geert-Jan A. Brummer**[2,3], **Mattia Greco**[1], **Lukas Jonkers**[1], **André Wizemann**[4], **Agnes K. M. Weiner**[1,5], **Kate Darling**[6,7], **Michael Siccha**[1], **Ronan Ledevin**[8], **Hiroshi Kitazato**[9], **Thibault de Garidel-Thoron**[10], **Colomban de Vargas**[11,12], **Michal Kucera**[1]

1 MARUM Center for Marine Environmental Sciences, University of Bremen, Leobener Strasse, Bremen, Germany, 2 NIOZ Royal Netherlands Institute for Sea Research, Department of Ocean Systems, and Utrecht University, Den Burg, and Utrecht University, The Netherlands, 3 Vrije Universiteit Amsterdam, Department of Earth Sciences, Faculty of Science, Amsterdam, The Netherlands, 4 Leibniz Centre for Tropical Marine Research, Bremen, Germany, 5 Department of Biological Sciences, Smith College, Northampton, Massachusetts, United States of America, 6 School of GeoSciences, University of Edinburgh, Edinburgh, Scotland, United Kingdom, 7 School of Geography and Sustainable Development, University of St Andrews, St Andrews, Scotland, United Kingdom, 8 UMR5199 PACEA, Université de Bordeaux, Allée Geoffroy Saint Hilaire, Pessac, France, 9 Japan Agency for Marine Earth Science and Technology (JAMSTEC), Yokosuka, Kanagawa, Japan, 10 Aix-Marseille Université, CNRS, IRD, Collège de France, INRA, CEREGE, Aix-en-Provence, France, 11 Sorbonne Université, CNRS, Station Biologique de Roscoff, UMR 7144, ECOMAP, Roscoff, France, 12 Research Federation for the Study of Global Ocean Systems Ecology and Evolution, FR2022/Tara GOSEE, Paris, France

* rmorard@marum.de

**Data Availability Statement:** All newly generated Sanger sequences are accessible on NCBI under the accession numbers MN383323-MN384218.

## Abstract

The planktonic foraminifera genus *Globigerinoides* provides a prime example of a species-rich genus in which genetic and morphological divergence are uncorrelated. To shed light on the evolutionary processes that lead to the present-day diversity of *Globigerinoides*, we investigated the genetic, ecological and morphological divergence of its constituent species. We assembled a global collection of single-cell barcode sequences and show that the genus consists of eight distinct genetic types organized in five extant morphospecies. Based on morphological evidence, we reassign the species *Globoturborotalita tenella* to *Globigerinoides* and amend *Globigerinoides ruber* by formally proposing two new subspecies, *G. ruber albus* n.subsp. and *G. ruber ruber* in order to express their subspecies level distinction and to replace the informal *G. ruber* "white" and *G. ruber* "pink", respectively. The genetic types within *G. ruber* and *Globigerinoides elongatus* show a combination of endemism and coexistence, with little evidence for ecological differentiation. CT-scanning and ontogeny analysis reveal that the diagnostic differences in adult morphologies could be explained by alterations of the ontogenetic trajectories towards final (reproductive) size. This indicates that heterochrony may have caused the observed decoupling between genetic and morphological diversification within the genus. We find little evidence for environmental forcing of either the genetic or the morphological diversification, which allude to biotic interactions such as symbiosis, as the driver of speciation in *Globigerinoides*.

**Funding:** This work was supported by grants from ANR-09-BLAN-0348 POSEIDON, ANR-JCJC06-0142-PALEO-CTD, from Natural Environment Research Council of the United Kingdom (NER/J/S2000/00860 and NE/D009707/1), the Leverhulme Trust and the Carnegie Trust for the Universities of Scotland, from DFG-Research Center/Cluster of Excellence 'The Ocean in the Earth System', from the Deutsche Forschungsgemeinschaft KU2259/19 and through the Cluster of Excellence "The Ocean Floor – Earth's Uncharted Interface".

**Competing interests:** The authors have declared that no competing interests exist.

# Introduction

Species of the genus *Globigerinoides* are the dominant constituent of tropical-subtropical planktonic foraminifera assemblages throughout the Neogene and represent a cornerstone for paleoceanography. The extant members of the genus feature one of the most iconic species of planktonic foraminifera that was formally described from the Atlantic by d'Orbigny in 1839 as *Globigerina rubra*, after the reddish coloration of its test. The species definition was later widened to include colorless specimens as variants with the same morphology because shell color was not considered taxonomically relevant at the species level [1,2]. It was further broadened by Parker [3] to include the morphologically similar *Globigerinoides elongatus* (d'Orbigny) and *Globigerinoides pyramidalis* (van den Broeck) that were originally distinguished using characteristics such as the compression of the last chamber and a higher trochospire. Parker [3] considered that the three species formed a morphological continuum with *G. ruber* and this broad definition was endorsed by Kennett and Srinivasan in 1983 [4], who interpreted *G. elongatus*, *G. pyramidalis* and also *G. cyclostomus* (Galloway and Wissler) as ecophenotypic variants of *G. ruber*. This broad species definition has remained stable since, but most researchers continued to distinguish the two "chromotypes" as *G. ruber* "white" and *G. ruber* "pink", because of differences in biogeography, seasonality and isotopic composition [5]. Their distinction is particularly highlighted by the extinction of *G. ruber* "pink" in the Indian and Pacific Oceans 120,000 years ago, while persisting in the Atlantic to the present day [6].

The lumping of *G. elongatus*, *G. pyramidalis*, *G. cyclostomus* with *G. ruber* was questioned by Robbins and Healy-Williams [7], who identified stable isotopic differences among morphological variants. This motivated Wang [8] to further test for isotopic differences between morphological variants of *G. ruber* "white". Wang [8] informally re-created the split between *G. ruber* and *G. elongatus*, that had already been identified by d'Orbigny and referred to the original *G. ruber* as *G. ruber* sensu stricto (s.s.) and lumped the specimens matching the description of *G. elongatus*, *G. pyramidalis* and *G. cyclostomus* into *G. ruber* sensu lato (s.l.). Wang [8] showed subtle but statistically significant differences of 0.21 ± 0.21‰ for $\delta^{18}$O and −0.28±0.29‰ for $\delta^{13}$C between the two informal taxonomic units in the South China Sea and suggested that *G. ruber* s.s. lived in the upper 30 meters of the water column and *G. ruber* s.l. lived below 30 meters. Wang used this feature to reconstruct the variation of the thermal structure of the water column during the last glacial cycle. The work of Wang [8] triggered a series of studies during the last two decades that examined chemical/compositional, morphological and ecological differences between *G. ruber* s.s. and *G. ruber* s.l. [8–21] to assess their usefulness for paleoceanography.

In parallel to the investigation of the ecology of *G. ruber* s.l. and s.s., sequencing of the small sub-unit of the ribosomal RNA gene (SSU rDNA) shed new light on the diversity within the genus *Globigerinoides*. The earliest molecular phylogenies by Darling et al. [22,23] demonstrated that the two chromotypes of *G. ruber* are genetically distinct, in line with the well-established biogeographical and ecological differences [5]. Later, Darling and Wade [24] described further genetic diversity within *G. ruber* "white" and Kuroyanagi et al. [25] suggested that the genetic discontinuity observed within *G. ruber* "white" mirrored the sensu stricto/sensu lato division of Wang [8]. These observations were confirmed by Aurahs et al. [26] who identified four genotypes in *G. ruber* "white" (Ia, Ib, IIa and IIb) and in a second study [27], these authors analysed images of the barcoded specimens to show that genotypes Ia and Ib matched the diagnosis of *G. ruber* s.s., whilst genotype IIa matched the diagnosis of *G. ruber* s.l. As a result, they proposed to reinstate *Globigerinoides elongatus* as a valid name for genotype IIa. Irrespective of the complicated taxonomy, all genetic studies consistently

identified *G. elongatus* (or *G. ruber* s.l.) as a sister to the morphologically distinct species *G. conglobatus*. The contrast between the genetic divergence and morphological similarity of *G. elongatus* and *G. ruber* implies a disconnection between genetic and morphological evolution in the genus. Thus, next to the need to clarify and stabilize its nomenclature, the complex diversification pattern in the genus also calls for a comprehensive study of the pattern of speciation and morphological diversification leading to the present-day diversity in *Globigerinoides*.

To this end, we assembled a global dataset of single-cell SSU rDNA sequences covering all morphospecies of the genus, applied an objective molecular nomenclature system [28] to parse the genetic variability and used the shell morphology of the barcoded specimens to map the genetic units onto a morphological taxonomic framework. To explore patterns of morphological evolution within the genus, we used CT scanning to quantify the ontogenetic trajectory of the five morphospecies [29,30]. This allowed us to investigate whether the diagnostic differences in adult morphology between closely related species in the genus could be the result of heterochrony, with slight alteration in the developmental sequence leading to large differences in adult shape and size. Finally, we use our collection of globally distributed samples to analyze the ecology of the morphological and cryptic species in the genus and discuss the potential drivers of their evolution.

## Material

Living planktonic foraminifera of the morphospecies *Globoturborotalita rubescens*, *Globigerinoides ruber*, *Globigerinoides conglobatus*, *Globigerinoides elongatus* and *Globigerinoides tenellus* were sampled between 1993 and 2015 during 23 research cruises and 6 near shore sampling campaigns (Fig 1) in all oceans. No sampling permit was needed for planktonic foraminifera. *G. rubescens* was included in the analysis to serve as outgroup in phylogenetic analyses. The specimens were sampled using different open-closing plankton net systems, simple plankton nets or ship pump systems between 0 and 700 m water depth and mesh sizes from 63 to 200 μm. The specimens were separated from other plankton, cleaned with brushes and either transferred onto cardboard slides and air-dried or directly transferred into DNA extraction buffer and stored at -20°C or -80°C. The specimens stored on cardboard slides were transferred into DNA extraction buffer later in the laboratory.

## Methods

### Molecular analyses

DNA extraction was performed using either the DOC protocol, the GITC* protocol or the Urea Protocol [31]. A fragment located at the 3'end of the of the SSU rDNA between the primers S14F1 or S14p and 1528R [32] was amplified and the PCR products obtained were purified and sequenced directly with Sanger sequencing by several service providers (LGC Genomics Berlin, University of Edinburgh Gene Pool, AGOWA and Station Biologique de Roscoff). In addition, we randomly selected eight specimens for cloning in order to quantify potential intragenomic variability and used the TOPO TA cloning kit (Invitrogen) according to manufacturer instructions. Between 2 and 13 clones were sequenced per individual. All chromatograms were carefully checked to ensure sequence quality and were deposited on NCBI under the accession numbers MN383323 to MN384218. The methodologies used for sampling, DNA extraction, amplification and cloning of single planktonic foraminifera cells are described in Weiner et al. [31].

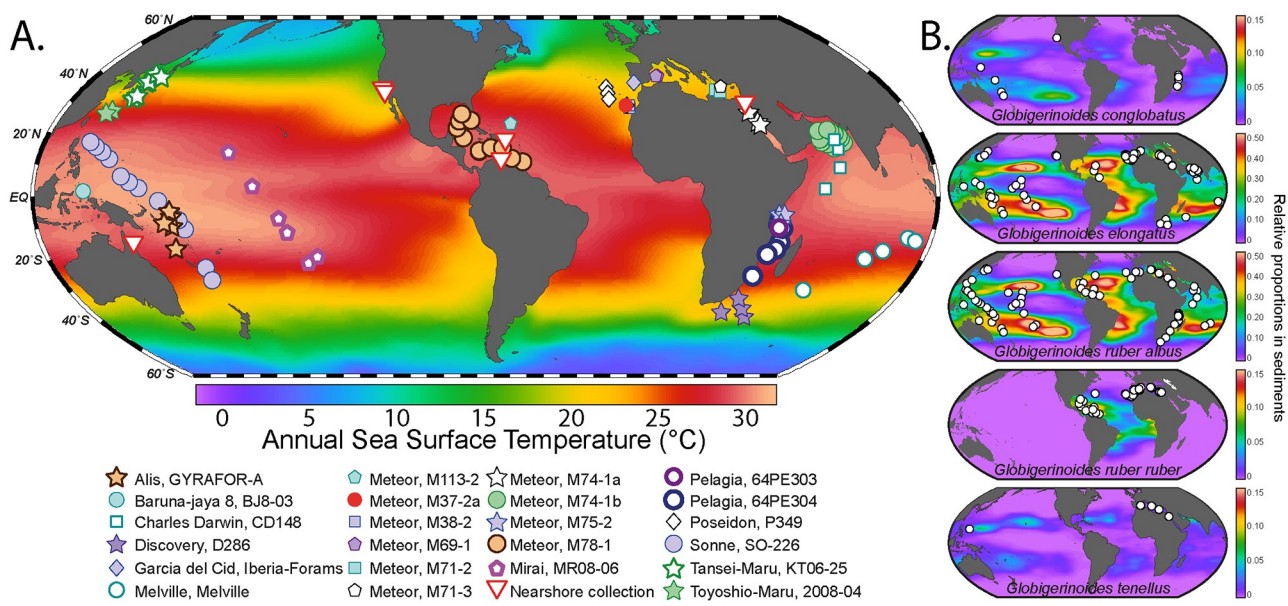

**Fig 1. Samples collection.** (A) Locations of the samples analyzed in the study. Each symbol corresponds to a scientific cruise or near shore collection site. Cruise names are indicated in the legend. The background color represent the annual sea surface temperature extracted from the World Ocean Atlas [105]. (B) Sampling coverage of the five species of the genus *Globigerinoides*. The colors in the background represent the relative abundance in sediments extracted from the FORCENS database [106]. Note that *G. ruber albus* n.subsp. and *G. elongatus* have the same map because they usually were not be discriminated in micropaleontological studies. The maps were generated using Ocean Data View [107].

## Public databases

We completed our dataset with sequences already made available by earlier studies. First, we retrieved all 359 SSU rDNA sequences of the six morphospecies that were stored in the PFR² database v 1.0 [33]. We then manually queried the NCBI portal (last accession: 15.11.2018) and retrieved seven additional sequences of *G. ruber* (Accession numbers KY397454-KY397460).

Detailed information on handling procedures, sequences and associated metadata of the newly generated data and those retrieved from public databases are provided in S1 Table.

## Molecular nomenclature

The genetic diversity within the six morphospecies was classified into a three-tier hierarchical scheme of Molecular Taxonomic Units following the system described in Morard et al [28]. The system uses the amplified ~1000 bp long sequence fragment located at the 3'end of the SSU rDNA between stems 32 and 50 as molecular marker [32], which is the barcode selected for benthic foraminifera [34] that covers six variable regions, three of which are foraminifera-specific. To exclude potential sequencing errors when constructing the nomenclature, we retained only sequences for which the individual sequence pattern was observed at least three times across our dataset. All distinct sequences in the resulting trimmed dataset were considered as *basetypes*. *Basetypes* co-occurring within one or several individuals (because of intra-individual variability among tandem copies of the gene) were assembled into *basegroups*, and constitute the lowest level of the nomenclature (MOTUs lvl-3). The variability observed between the basetypes represents at least the intragenomic (intra-individual) variability and the variability observed among different basegroups is considered to represent at least the level of population variability. If a unique basetype is observed within a single specimen, which is the majority of cases in our dataset (see Results), the resulting basegroup contains a single

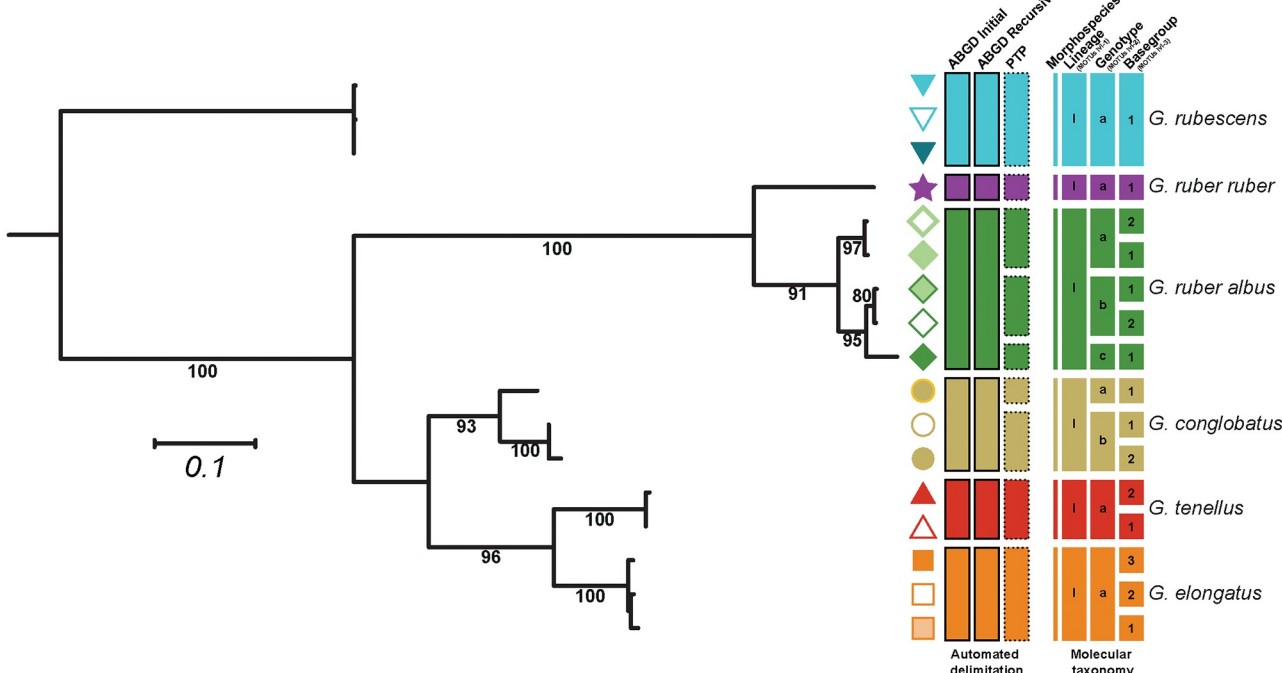

**Fig 2. Molecular taxonomy of the genus *Globigerinoides*.** Each branch represents a unique basetype, the symbol next to the branch represent the individual basegroup and the colors represent unique morphospecies. The first set of rectangles represent the three automated delimitation proposed by ABGD and PTP. The coarsest partition is retained as Lineage (MOTUs level-1) and encircled with a solid line, the finest partition is retained as Genotype (MOTUs level-2) and encircled in dotted line. The resulting 3-rank molecular taxonomy is showed in the second set of rectangles.

basetype. The levels 1 and 2 of the nomenclature (following Morard et al. [28]) were constructed using a combination of two automated delimitation methods, the Automated Barcode Gap Discovery method (ABGD; [35]) and the Poisson Tree Process (PTP; [36]). The sequences were aligned with MAFFT v.7 [37] and a phylogenetic inference was calculated with 1000 non-parametric bootstrapping pseudo replicates based on a BioNJ starting tree using PhyML [38]. The best substitution models were selected using the Smart Model Selection [39] under Akaike Information Criterion and the model GTR+I+G was selected. The resulting trees were submitted to the PTP server (http://species.h-its.org/) under default settings. The same alignment that served to generate the tree was submitted to the online ABGD server (http://wwwabi.snv. jussieu.fr/public/abgd/abgdweb.html) using the Kimura K80 distance and default options. We retained the initial (coarsest delimitation) and recursive partition (finest delimitation) provided with the lowest prior intraspecific divergence. We defined the MOTU lvl-2 as the finest delimitation proposed by either ABGD or PTP and the MOTU lvl-1 as the coarsest. The proposed delimitations are retained as working hypotheses provided that two clones belonging to the same basegroup were not attributed to different partitions (oversplit) and that sequences belonging to different morphospecies were not grouped in the same partition (lumping). The delimitation proposed by ABGD and PTP as well as the retained delimitation are reported in Fig 2. As multiple, but partly overlapping, nomenclatural schemes were proposed by successive studies [21–24, 40, 41], we reported the correspondence between these schemes and their equivalent in our system (Fig 3 and S2 Table).

A significant part of the sequences had insufficient quality and/or coverage to be included in the assessment of the diversity within the *Globigerinoides* plexus, but carried enough information to be attributed to at least one MOTU level of our nomenclatural system. The Sanger

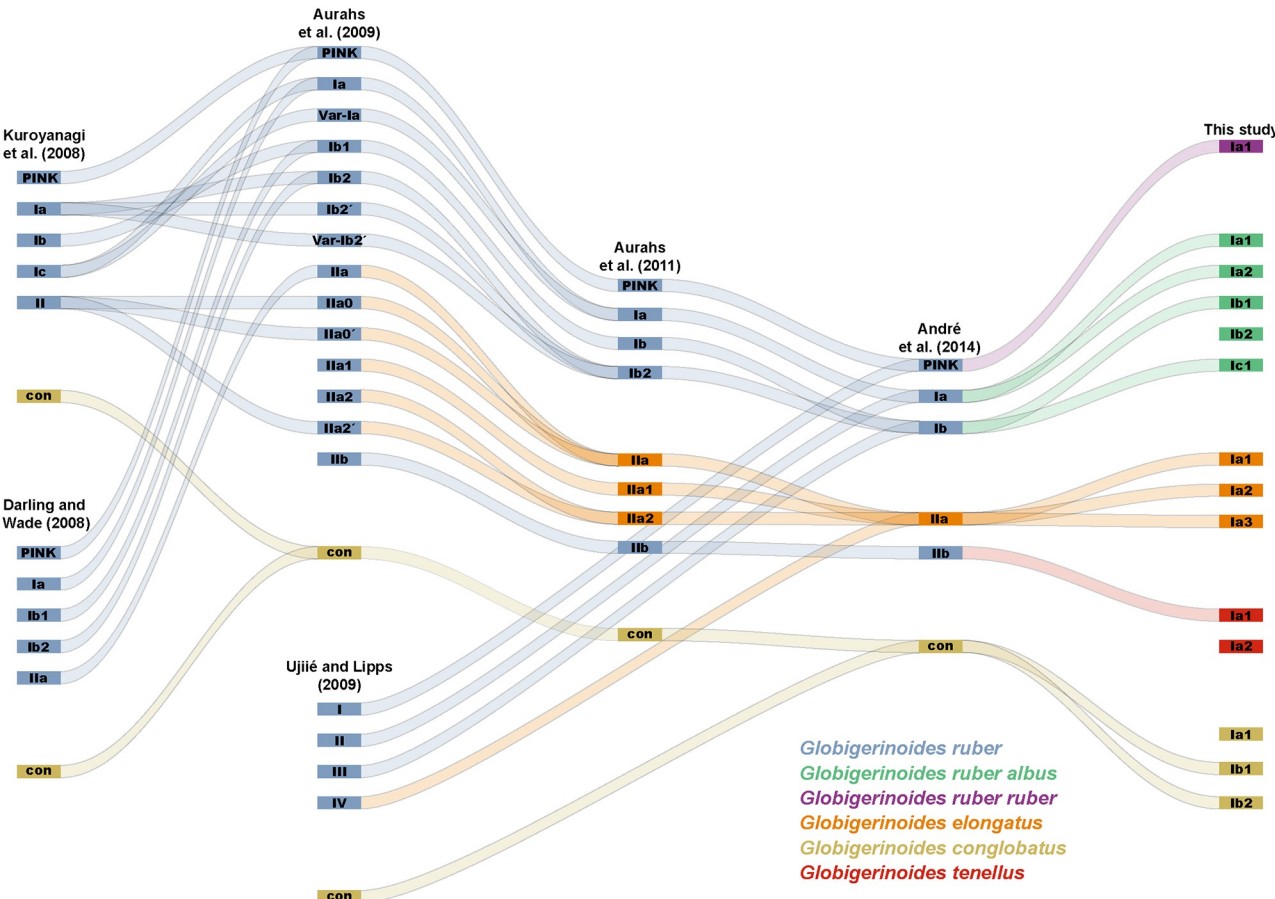

**Fig 3. Development and consistency across the nomenclatural scheme proposed for the genus *Globigerinoides*.** The Sankey diagram indicates the change in the names, addition of new taxa, lumping and splitting of existing units across the successive studies. The change of colors indicates when formal taxonomic revisions were made.

sequences not meeting the quality criteria were compared to the basetype sequences and received the finest taxonomic attribution possible based on the availability of diagnostic sites in the region they covered (See S1 Table). Biogeography and temporal occurrences of the genotypes and basegroups are shown in Fig 4.

## Sample coverage and environmental parameters

We calculated rarefaction curves at MOTUs lvl-2 and lvl-3 (Fig 5) and complemented the approach with a first order Jackknifing to evaluate the coverage of our dataset (Table 1). Because *G. tenellus* and *G. conglobatus* were under-sampled (60 sequences in a dataset of 1251 sequences), we calculated the rarefaction curves to include all species and selectively only for *G. ruber* and *G. elongatus* separately (Fig 5). Likewise, the Jackknifing was applied to *G. ruber* and *G. elongatus* combined at the MOTUs lvl-2 and on each species separately at the lvl-3 (Table 1). We then applied the analyses to the global dataset and separately on three main biogeographic regions: North Atlantic Ocean, Indian Ocean and Pacific Ocean.

The dataset constituted for this study is the result of the efforts by multiple research teams and re-exploitation of public data, therefore it was difficult to recover and harmonize the environmental parameters measured during each sampling campaign. In order to analyze the

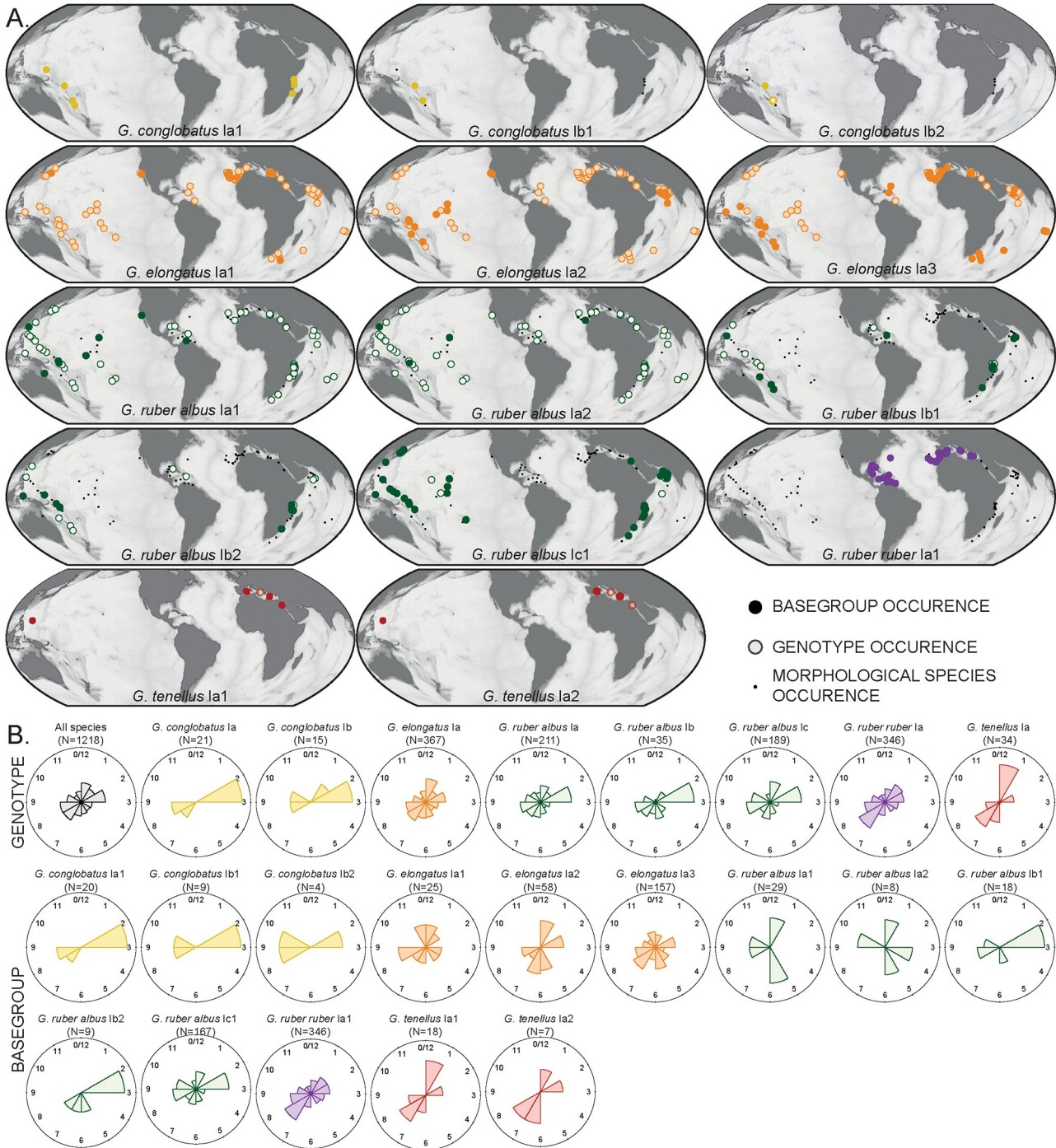

**Fig 4. Biogeographic distribution of constitutive genotypes (MOTUs lvl-2) and basegroups (MOTUs lvl-3) of the genus *Globigerinoides* in the sample set.** (A) The circles indicate where the genotypes have been collected and are filled when the basegroup has been identified in the sample. Note that the coverage for *G. conglobatus* and *G. tenellus* is insufficient for robust interpretation. The maps were generated using Ocean Data View [107]. (B) Windrose diagram showing the month of collection of each genotype and basegroup. The month of collection have been normalized in regard to hemisphere.

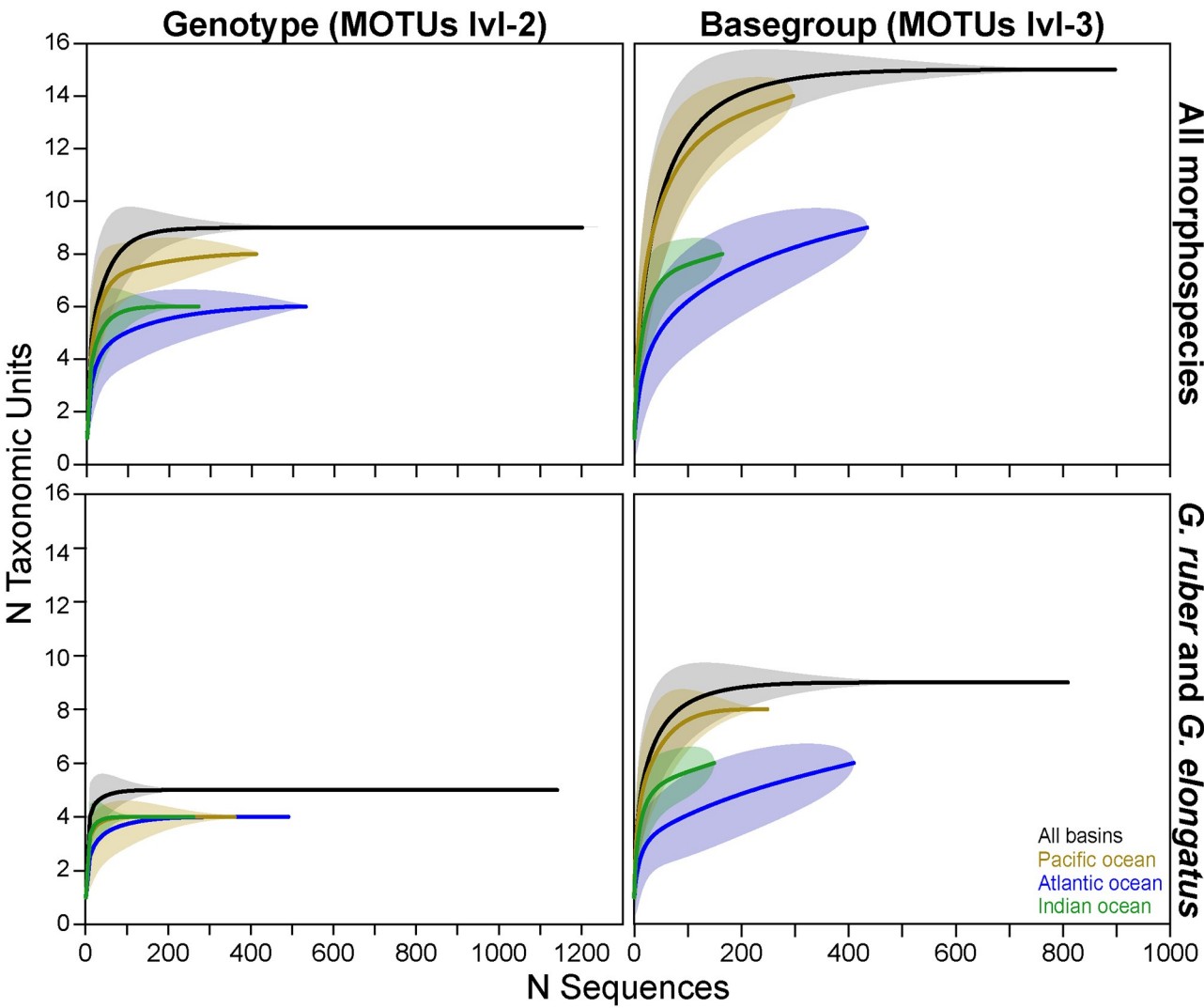

**Fig 5. Assessment of species richness.** Rarefaction curves for the different basins and the entire dataset at the genotype (MOTUs lvl-2) and basegroup (MOTUs lvl-3) levels, and for all morphospecies together and for the better sampled *G. ruber* and *G. elongatus* only.

ecological preferences of the sampled genotypes and basegroups, we chose to use geographic coordinates and collection date to extract the monthly average values of the following environmental parameters from public databases: Sea Surface Temperature (SST), Mixed Layer Salinity (MLS), Chlorophyll concentration (CHL), Particulate Organic Carbon (POC) and Productivity (PROD). The SST, CHL and POC parameters were extracted from the MODIS-Aqua (NASA, Greenbelt, MD, USA) database [42–44], the MLS was extracted from the Isopycnal/Mixed-layer Ocean Climatology (MIMOC) database [45] and PROD was calculated following the Vertically Generalized Production Model from Behrenfeld and Falkowski [46]. In this way, we could gather a homogeneous environmental dataset although it is less precise than in-situ measurements. We display the environmental parameter values at the morphospecies, genotype and basegroup levels in Fig 6 and tested if the distribution of values of sister taxa at each taxonomic level was the same (null hypothesis) with a simple non-parametric Wilcoxon-Mann-Whitney U-test using the Bonferroni correction (Table 2). All statistical analyses were performed in PAST 3.21 [47].

**Table 1. Results of the Jackknifing analyses that provide the comparison between the observed diversity ($S_o$) and the estimated basegroup diversity ($S_e$) for *G. ruber* and *G. elongatus* basegroup at global and basins scales.** Note that the entire diversity of *G. ruber* and *G. elongatus* may not have been entirely captured in the Atlantic Ocean and the Indian Ocean respectively because $S_o$ does not fall into the 95% confidence interval ($CI_{95}$).

| | | Global | North Atlantic Ocean | Indian Ocean | Pacific Ocean |
|---|---|---|---|---|---|
| *G. ruber (albus + ruber)* + *G. elongatus* (GENOTYPE) | $S_o$ | 5 | 4 | 4 | 4 |
| | $S_e$ | 5 | 4 | 4 | 4 |
| | $CI_{95}$ | 0 | 0 | 0 | 0 |
| | So $\in$ Se $\pm$ CI95 | TRUE | TRUE | TRUE | TRUE |
| *G. ruber (albus + ruber)* + *G. elongatus* (BASEGROUP) | $S_o$ | 9 | 6 | 6 | 8 |
| | $S_e$ | 9 | 7.97183 | 6.97872 | 8 |
| | $CI_{95}$ | 0 | 2.713228 | 1.9182971 | 0 |
| | So $\in$ Se $\pm$ CI95 | TRUE | FALSE | FALSE | TRUE |
| *G. ruber (albus + ruber)* (BASEGROUP) | $S_o$ | 6 | 4 | 3 | 5 |
| | $S_e$ | 6 | 5.97183 | 3 | 5 |
| | $CI_{95}$ | 0 | 2.713228 | 0 | 0 |
| | So $\in$ Se $\pm$ CI95 | TRUE | FALSE | TRUE | TRUE |
| *G. elongatus* (BASEGROUP) | $S_o$ | 3 | 2 | 3 | 3 |
| | $S_e$ | 3 | 2 | 3.97872 | 3 |
| | $CI_{95}$ | 0 | 0 | 1.9182971 | 0 |
| | So $\in$ Se $\pm$ CI95 | TRUE | TRUE | FALSE | TRUE |

## Phylogeny and molecular clock

To reconstruct the evolutionary history of the genus *Globigerinoides*, we applied a molecular clock estimation using the same alignment as for the maximum likelihood tree inference (Fig 2). We used the divergence between *G. rubescens* and the genus *Globigerinoides* (23.8 Ma [48]), the First Appearance Datum (FAD) of *G. conglobatus* (8–8.6 Ma) and *G. tenellus* (2.5 Ma), which are known from the fossil record [49], as minimum ages to constrain the phylogeny. We used a relaxed clock model implemented in BEAST v.1.8.4 [50]. Model parameters were set using BEAUti v1.8.4. The distribution of the fixed node age prior was considered normal and the speciation rate was assumed constant under the Yule-Process. The GTR (Generalised Time Reversible) model was selected as substitution model and an UPGMA (Unweighted Pair Group method with arithmetic mean) tree was calculated as starting tree. Markov-Chain-Monte Carlo (MCMC) analyses were conducted for 10,000,000 generations, with a burn-in of 1000 generations and saving each 1000th generation. The maximum clade credibility tree with median node heights was calculated in TREEAnnotator from the BEAST package, with a burn-in of 100 trees and a posterior probability limit of 0. The resulting tree was then visualized in FigTree v. 1.3.1 [51] and is shown in Fig 7.

## 3D morphology

We produced CT-scans of *G. rubescens*, *G. ruber albus* n.subsp., *G. conglobatus*, *G. elongatus* and *G. tenellus* to assess the ontogenetic development of each species. To ensure that the specimens had completed their life cycle, which usually is not the case for the living specimens collected in the water column, we used specimens recently deposited on the seafloor from a core top sample retrieved south of Barbados at station GeoB3935 (12˚36.8 N, 59˚23.2 W; bottom depth 1554 meters) [52]. We chose this sample because of the exceptional preservation of the tests, which were free of fine-grained sediment. Moreover, its provenance is close to the sampling localitions where *Globigerinoides* spp. were previously analysed for their ontogeny [53]. From this sample, we selected one specimen per morphospecies, choosing specimens with

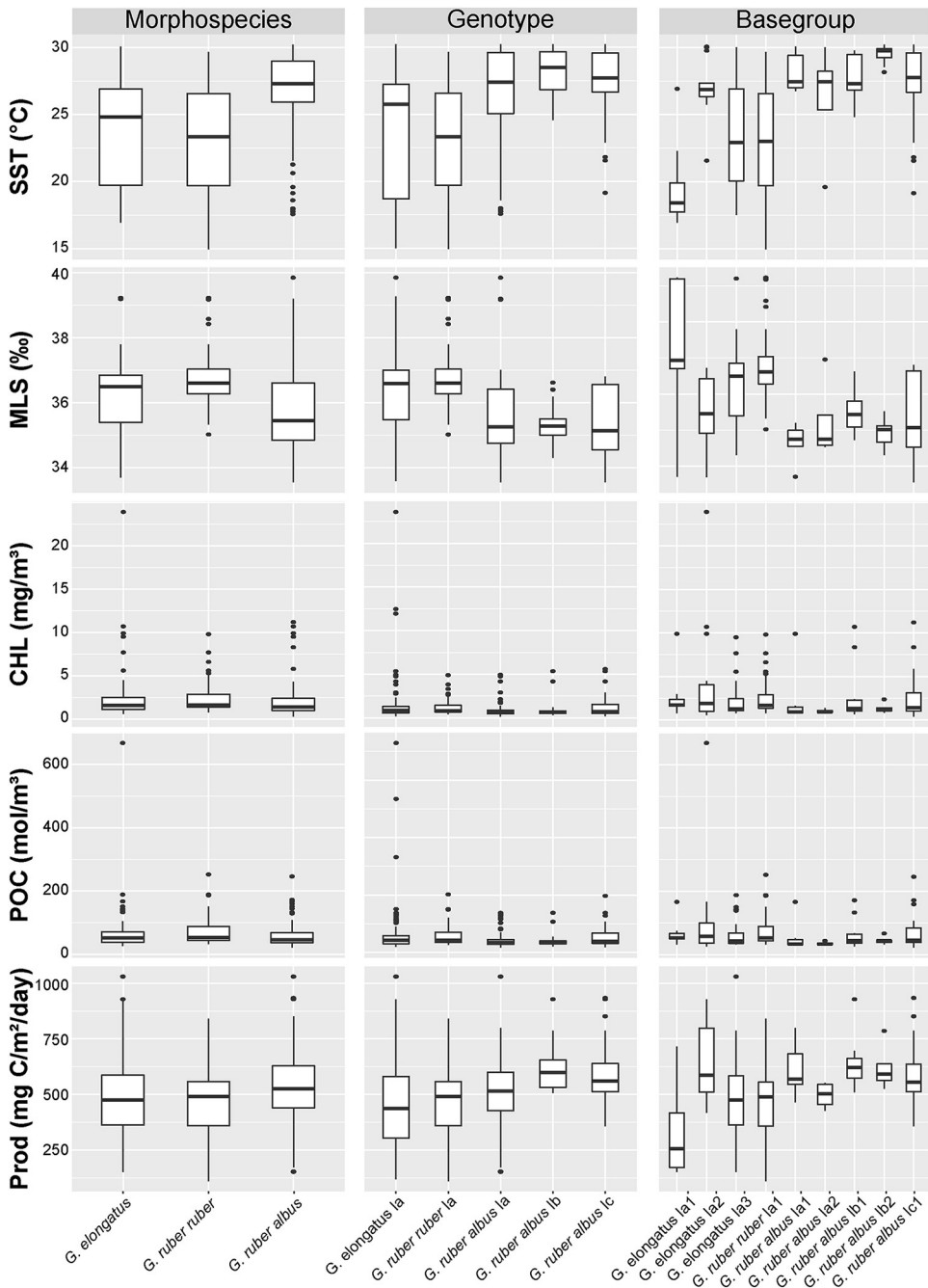

**Fig 6. Environmental parameters.** Distribution of the monthly values of Sea Surface Temperature (SST), Mixed Layer Salinity (MLS), Chlorophyll (CHL), Particulate Organic Carbon (POC) and Productivity (Prod), observed for the morphospecies, genotypes and basegroups of *G. elongatus*, *G. ruber albus* n.subsp. and *G. ruber ruber*. The statistical tests to compare the distribution are provided in Table 2. The box plot were generated with R [108] using the ggplot2 package [109].

well-developed characteristic features. We choose specimens that were of large size and had a thick test, which indicates maturity and facilitates CT-scanning at good resolution. Indeed, four of the five species have a diminutive final chamber indicative of reproduction by gametogenesis (the reproductive terminal stage sensu Brummer et al. [53]), while *G. ruber* is

**Table 2. Results of Mann-Whitney tests for environmental parameters comparisons.** The significant values are shown in bold.

| Morphospecies | SST | MLS | CHL | POC | Prod |
|---|---|---|---|---|---|
| *G. elongatus* vs *G.ruber albus* | **1.18E-04** | **0.02** | 0.82 | 1.00 | 0.18 |
| *G. elongatus* vs *G.ruber ruber* | 0.72 | 0.10 | 0.87 | 1.00 | 1.00 |
| *G.ruber albus* vs *G.ruber ruber* | **2.32E-07** | **3.82E-07** | 0.05 | 0.20 | **0.01** |
| **Genotype** | | | | | |
| *G. ruber albus* Ia vs *G. ruber albus* Ib | 1.00 | 1.00 | 1.00 | 1.00 | **0.02** |
| *G. ruber albus* Ia vs *G. ruber albus* Ic | 1.00 | 1.00 | 1.00 | 1.00 | 0.29 |
| *G. ruber albus* Ib vs *G. ruber albus* Ic | 1.00 | 1.00 | 1.00 | 1.00 | 1.00 |
| **Basegroup** | | | | | |
| *G. elongatus* Ia1 vs *G. elongatus* Ia2 | **2.00E-03** | **0.02** | 1.00 | 1.00 | **0.01** |
| *G. elongatus* Ia1 vs *G. elongatus* Ia3 | 0.10 | 0.43 | 1.00 | 1.00 | 0.25 |
| *G. elongatus* Ia2 vs *G. elongatus* Ia3 | 0.56 | 1.00 | 1.00 | 1.00 | 0.37 |
| *G. ruber albus* Ia1 vs *G. ruber albus* Ia2 | 1.00 | 1.00 | 1.00 | 1.00 | 1.00 |
| *G. ruber albus* Ib1 vs *G. ruber albus* Ib2 | 0.28 | 0.82 | 1.00 | 1.00 | 1.00 |

normalform. We realize that planktonic foraminifera are morphologically variable, not only in their adult shape but also throughout their ontogeny [53], so the decision to analyze only a single specimen per morphospecies was made in order to achieve a first rough assessment of the main differences of ontogenetic trajectories among the morphospecies. Such trajectories are known to differ between species but are stable within species, with much variability correlated with proloculus size [29,30,53]. The selected specimens were individually mounted on a stub and scanned at a cubic resolution of 1.2 μm with a General Electrics V/Tome/x micro-scanner (PACEA, Bordeaux University). Each scan was performed at 80 kV and 180 μA without filter as the shell had a low X-ray absorption rate. The smaller specimen of *G. rubescens* was analyzed with a cubic resolution of 0.68 μm with a Zeiss Versa 500 at 80kV, 7W and with a filter LE1.

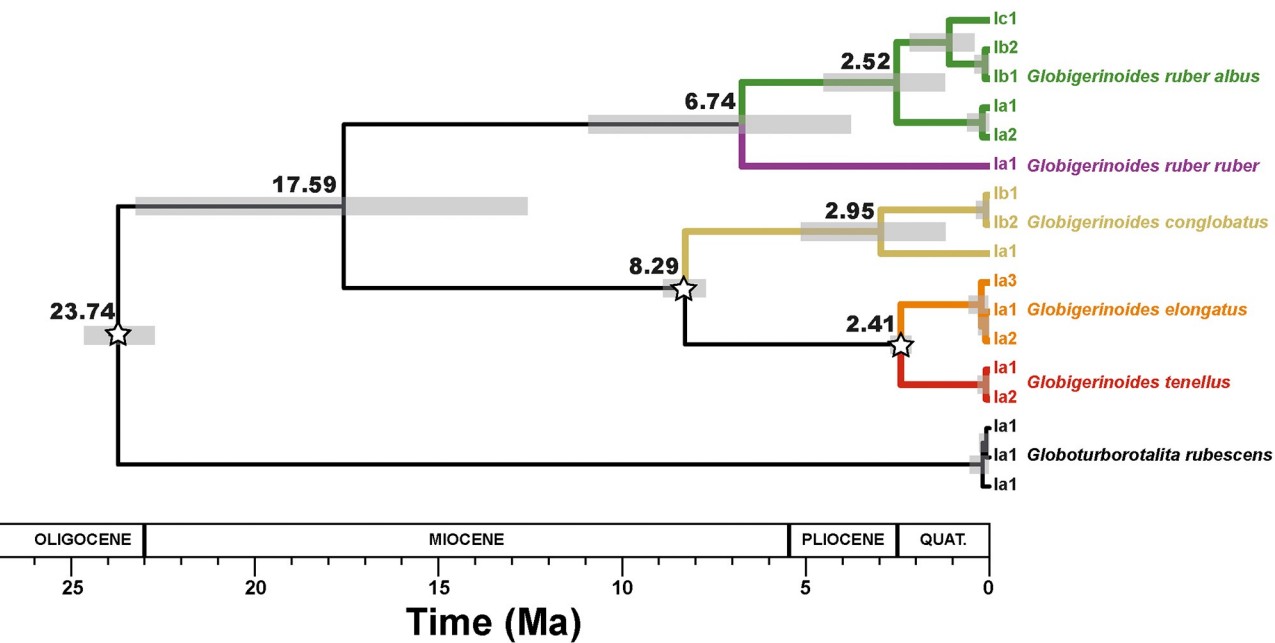

**Fig 7. Molecular clock estimates of the diversification of the *Globigerinoides* genus rooted on *Globoturborotalita rubescens*.** The grey bars indicate the uncertainties in the dating of the node and the stars indicate the nodes used for calibration (See text for details).

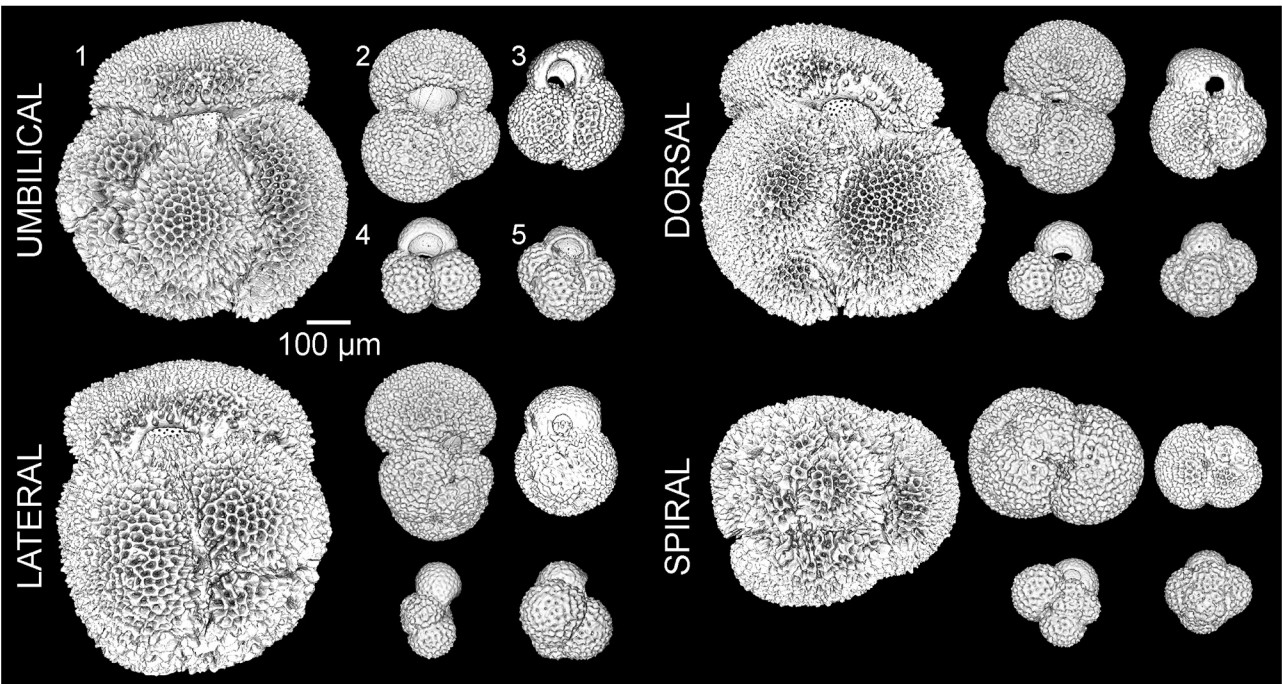

**Fig 8. 3D morphology.** CT- scans of external morphology of representative specimens of the five species in four standard views for (1) *G. conglobatus*, (2) *G. ruber*, (3) *G. elongatus*, (4) *G. tenellus* and (5) *G. rubescens*. The scaling of the species respects the difference in sizes.

Semi-automated segmentation was used to reconstruct three-dimensional (3D) virtual surfaces of the calcite volume (external morphology) of each specimen (Fig 8), and the inner volume of individual chambers (Fig 9) were produced by manual segmentation with the ITK-SNAP v 3.6 software [54] to reconstruct the ontogenetic trajectory of each morphospecies. We automatically extracted the volume, centroid position and major axis of individual chambers using a custom script in MATLAB R2017b to calculate growth parameters of the trochospire, following the model of Raup [55]. We calculated the whorl expansion rate *W*, the relative distance between the generating curve and the axis of coiling *D*, the translation rate *T* and the shape of the generating curve *S*. The calculated growth parameters of each species are displayed in Fig 10 and the numerical values are provided in S3 Table.

## Results

### Genetic diversity within *Globigerinoides*

Our dataset on the molecular diversity within the genus *Globigerinoides* and its sister species *G. rubescens* includes 1251 Sanger sequences, of which 893 are new. All 1251 sequences cover the same rDNA barcode region and originated from a total of 1159 individuals collected at 179 sampling stations (Fig 1). Among the 1251 sequences, 147 met the quality criteria to derive molecular taxonomy and served to define a total of 17 basetypes (unique, replicable sequence motifs). We observed three basetypes that co-occurred within two single individuals of *G. rubescens* that were consequently grouped into a single basegroup. Additionally, we identified the co-occurrence of two basetypes within three clones from a single individual of *G. ruber*, published by Kuroyanagi et al. [25]. Since this is the only observation of intragenomic variability within the SSU rRNA gene in *G. ruber*, we consider it likely that it resulted from contamination or PCR/sequencing error and we thus reject this single observation as evidence

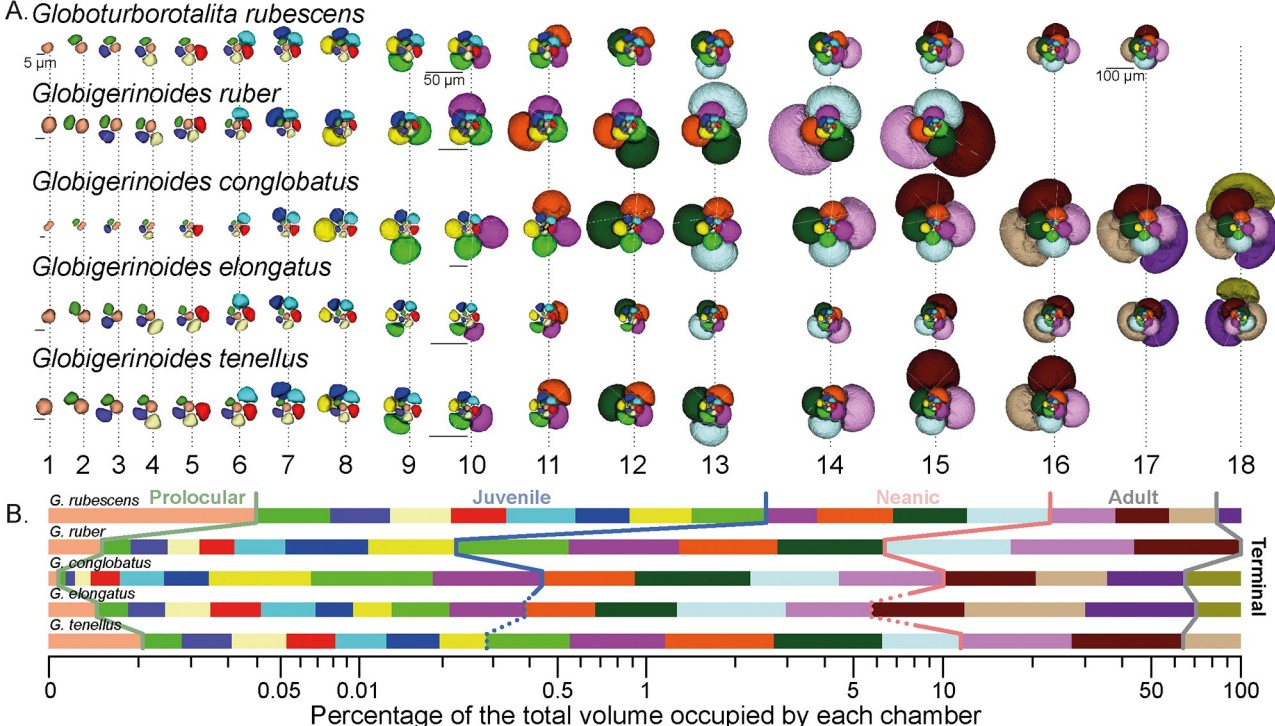

**Fig 9. Ontogenetic development of the five selected morphospecies.** (A) The addition of individual chambers is shown with segmentation of the inner volume from the proloculus to the final chamber. To accommodate the difference in size during the ontogeny and between the species, we have decreased the relative size of the successive stage by 10% and provide scale bars at the beginning, middle and end of their ontogeny for reference. (B) Relative proportions of the total inner volume occupied by each chamber. Color coding of the chamber is the same as in (A) with indication of the transition between the successive ontogenetic stages marked colored lines (See main text for details). The dotted lines indicate when the exact transition between stages is uncertain.

for intragenomic variability in the species. As a result, we retained 15 basegroups (Fig 2), 14 of these consisting of a single basetype, which provided a basis for the construction of a molecular nomenclature of the group. The automated taxa partitions proposed by ABGD and PTP did not violate any of the conditions of the taxonomic system (lumping of sequences belonging to different morphotaxa or splitting of basetypes belonging to the same basegroup) and were thus retained. Partitions by ABGD reflected the morphological species concept of the group. The PTP analysis identified three partitions within *G. ruber albus* n.subsp. and two within *G. conglobatus*, which were retained as distinct genotypes. No partitions were identified within the morphospecies *G. rubescens*, *G. elongatus*, *G. tenellus* and *G. ruber ruber* indicating that these morphospecies consist of only a single genotype.

## Molecular and morphological revision of existing taxonomic concepts

The first DNA sequences of members of the genus *Globigerinoides* were made available in the earliest publications on the genetic diversity of planktonic foraminifera [23,56–58], but nomenclatural schemes to describe the cryptic diversity in the genus were presented only a decade later in parallel and independently by Darling and Wade [24] and Kuroyanagi et al. [25], who both identified five cryptic species within *G. ruber* (Fig 3). The complexity of naming cryptic species further increased in the following year when Ujiié and Lipps [40] produced a distinct nomenclatural scheme with only four cryptic species within *G. ruber*, whereas Aurahs et al. [26] further developed the scheme initially proposed by Darling and Wade [24], but

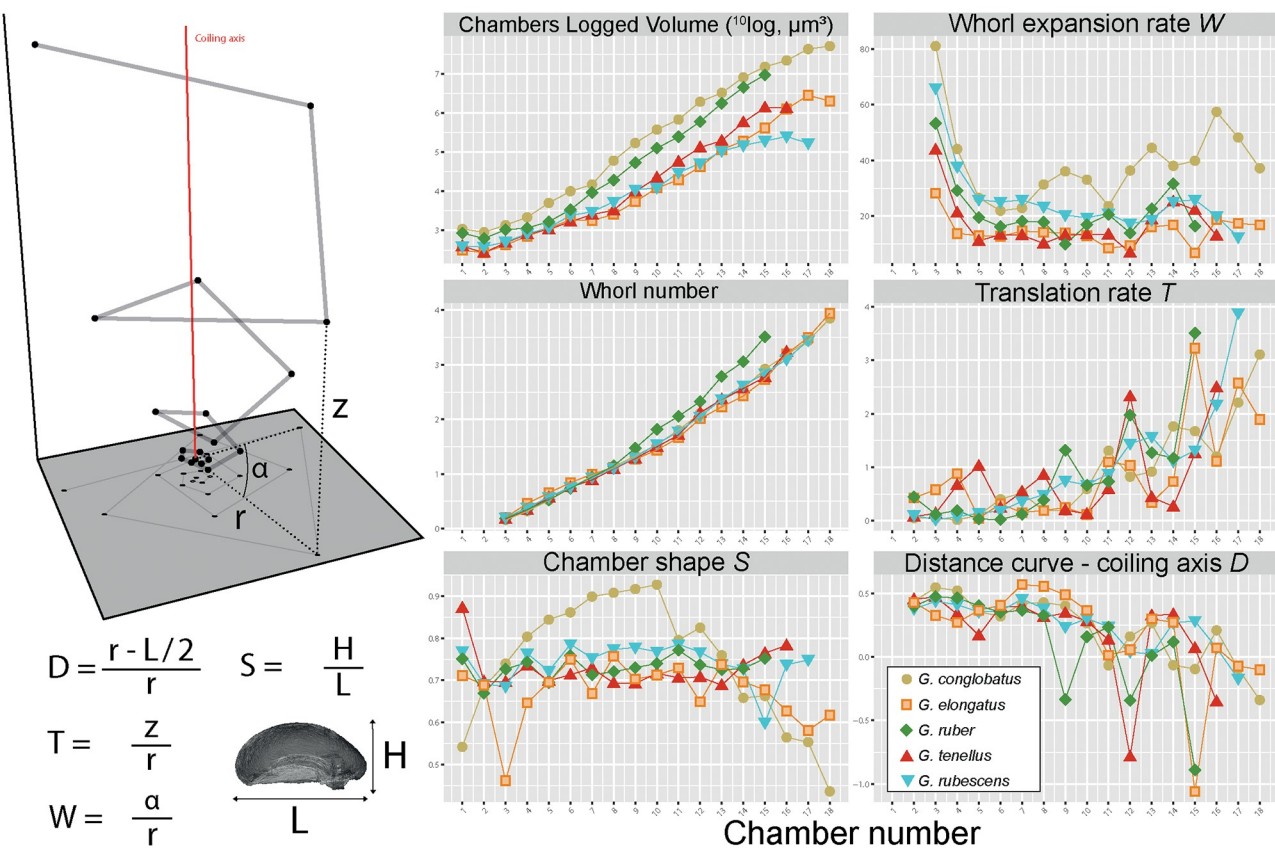

**Fig 10. Raup's parameters.** The scheme on the left represents the position of the centroids of the chambers in *G. conglobatus* in 3D space. The z-axis is given by the coiling axis of the specimen. The radius *r* (distance between the coiling axis and the centroid of a given chamber), the height *z* (distance between the centroids of the proloculus and a given chamber along the coiling axis) and the angle α (measured between the radii of two successive chambers) are illustrated on the scheme. The segmentation of the inner volume of the last chamber is given in the right bottom corner of the scheme together with the biometric measures H (Height of the chamber) and L (Length of the chamber). The equations of the parameters of the Raup model are provided next to the graph (See explanation in the main text). The six panels on the right show the results for the Raup parameters for each chamber of each specimen together with the cumulative volume and the whorl number. The results of the measurements and calculation of the Raup parameters are provided in the S3 Table.

chose to interpret all subtle sequence differences across their dataset and produced a scheme with 14 different cryptic species. Two years later, Aurahs et al. [27] reduced the diversity to only eight cryptic species by considering only the most repeatable sequence pattern in their dataset. Furthermore, they split the genetic diversity between *G. ruber* s. s. (*G. ruber* Ia, Ib, Ib2 and pink) and *G. ruber* s. l. (*G. ruber* IIa, IIa1, IIa2 and IIb) with the genotype IIa being considered as *G. elongatus*. André et al. [59] proposed a revision of the available nomenclature based on automated methods for species delimitation to define cryptic diversity in planktonic foraminifera, which reduced the diversity to five cryptic species only.

In our study, we quadrupled the size of the dataset compared to previous studies and placed all the formally described cryptic species into a new framework, and we identified only one new basegroup in *G. ruber albus* n.subsp. (Ib2). We have also generated a SSU rDNA sequence from a specimen identified on collection and by later observations as *G. tenellus* (S1 Fig) that was identical to the sequences obtained from specimens of *G. ruber* Type IIb of Aurahs et al. [27]. This allowed us to recognize this type as *G. tenellus* and thus return the species to *Globigerinoides*, as a sister to *G. elongatus*. The extended and strictly curated dataset allowed for

identifying one new basegroup in *G. conglobatus* as well as in *G. tenellus* and reducing the number of basetypes to three within *G. elongatus.*

Since the genetic distance separating the "pink" and "white" chromospecies of *G. ruber* is greater than the distance separating *G. elongatus* and *G. tenellus* (Fig 2), we feel compelled to express the genetic and phenotypic distinction between the two lineages in formal taxonomy. However, the phenotypic distinction reflects the color of the shell, not shell morphology, and this character fades with age, rendering it impossible to distinguish the lineages in fossil material older than ~750 kyr [6]. Therefore, we propose to use the subspecies names *G. ruber albus* n.subsp. and *G. ruber ruber*, facilitating continuity by allowing the use of the nominotype "*G. ruber*" at the species level in situations where the chromospecies cannot be differentiated. Moreover, the physical holotype designated here combines the shell morphology with the SSU rDNA drawn from the same individual, the first for a planktonic foraminifer. The physical specimens have been deposited at the Naturalis Biodiversity Center, Leiden, the Netherlands.

## Systematics

Phylum Foraminifera d'Orbigny, 1826

 Class Globothalamea Pawlowski, Holzmann & Tyszka, 2013

 Order Rotaliida Delage and Hérouard, 1896

 Superfamily Globigerinoidea Carpenter, Parker & Jones, 1862

 Family Globigerinidae Carpenter, Parker & Jones, 1862

 Genus *Globigerinoides* Cushman, 1927, amended by Spezzaferri et al., 2015

 Type species *Globigerina rubra* d'Orbigny, 1839

 Species *Globigerinoides ruber* (d'Orbigny, 1839)

 Subspecies *Globigerinoides ruber albus* n. subsp.

Type material: Holotype: Voucher C319 collected at 7.409˚S, 165.274˚E on 12.03.2013 between 0–20 meters water depth (Museum number: RGM.1332320). Paratypes: Voucher C208 collected at 6.414˚N, 143.024˚E on 18.03.2013 between 80–100 meters water depth (Museum Number: RGM.1332321), Voucher C281 collected at 22.719˚S, 170.918˚E on 08.03.2013 between 60–80 meters water depth (Museum Number: RGM.1332322) and Vouchers C329 collected at 7.409˚S, 165.274˚E on 12.03.2013 between 0–20 meters water depth (Museum number: RGM.133233). Light microscopy images of the type specimens are provided in S2 Fig.

Diagnosis: Differs from *G. ruber ruber* by the absence of reddish color of the shell, by the presence of a distinct sequence motive in the SSU rDNA gene, by its seasonality and depth habitat in the modern Atlantic and its presence in the Indopacific throughout the last 120 ka. The two subspecies cannot be distinguished prior to 750 ka due to the fading of the color with time and both are then captured as *G. ruber* well into the Neogene.

Description. The new subspecies largely overlaps with *G. ruber ruber* in test morphology, but differs in the color of the test, which develops during the neanic stage [53]. The morphology of the species and its changes during the ontogeny have been described in detail by Brummer et al. [53] and is formalized accordingly below. The holotype has been selected such that the test shows all key features of the species, but lacks color and because it yielded a SSU rDNA sequence of genetic type *G. ruber albus* n.subsp. Ia (Voucher C319).

Prolocular stage. Proloculus small, 12.5 ± 1.5 μm (10–16 μm), wall imperforate, smooth and non-spinose; aperture interiomarginal, circular with thickened rim, in multi-chambered tests larger than deuteroconch and truncated by flat wall shared with deuteroconch.

Juvenile stage. Starting with deuteroconch, test lobate, umbilico-convex, umbilicus open, wide, narrowing after completion of initial whorl; chambers hemispherical, 7–12 (9.3 ± 1.2)

added in ± 1.5 whorls of near planispire, with 5–6 in initial whorl, totaling 8–13 (9.7 ± 1.2) chambers in tests 54–76 (65.3 ± 5.8) μm in diameter. Aperture interiomarginal-marginal, a small, low arch with marked rim. Spines sparse, thin, flexible; microspines present; pores sparse, exclusively along sutures on spiral side; wall texture spinose, non-cancellate. No preferential shell coiling direction; algal symbionts acquired.

Neanic stage. Test rapidly changing towards adult morphology, becoming sphaeroidal with umbilicus closing; chambers globose, 3–4 in half to complete whorl of low trochospire, decreasing to 3 in last whorl, totaling 12–16 (14 ± 1.3) chambers in tests 120–190 (140 ± 25) μm in diameter. Aperture widening to a wide, high arch and migrating to the umbilicus. Spines and pores becoming numerous and evenly distributed; spines becoming thicker and more rigid; spine bases, inter-spine ridges and pore pits develop; wall becoming coarsely perforate and cancellate.

Adult stage. Test sphaeroidal to elongate with reddish color, chambers globose in a low-medium trochospire, at least 1, usually 2 to 3, up to 4 chambers are added, totaling 14–18 chambers in test >180, up to 510 μm in diameter, until reproduction (gametogenesis). Secondary aperture(s) develop. Wall texture cancellate-spinose and macroperforate.

Terminal stage. Usually one, occasionally two normalform and/or diminutive (kummerform) chambers are added, rarely one or two bullate chambers capping the secondary apertures. Spines progressively shed, wall coarsely perforate, smooth to coarsely cancellate. Loss of algal symbionts, loss of buoyancy. Terminal shells 230–560 μm in diameter with 15–19 chambers in 3–4 whorls of low to medium trochospire.

## Distribution and ecological preferences of *Globigerinoides* MOTUs

Although our study benefits from a globally distributed sampling, we unfortunately lack sampling points in the Southern Atlantic. The rarefaction curves, however, confirm that the genotype diversity within *Globigerinoides* likely has been entirely captured by our global dataset as well as in the individual ocean basins when considering all morphospecies and the better sampled *G. ruber ruber*, *G. ruber albus* n.subsp. and *G. elongatus* respectively (Fig 5). We are confident that all existing genotypes and the majority of basegroups have been detected, so that we are able to interpret their biogeographic patterns (Fig 4A). We observe that the genotypes *G. ruber albus* n.subsp. Ia and Ib are cosmopolitan whilst the genotype *G. ruber albus* n.subsp. Ic was not found in the North Atlantic. A similar pattern could hold for the basegroup *G. ruber albus* n.subsp. Ib2 as well, as it has not been found in the North Atlantic. This may be a sampling bias because its genotype has been encountered only at two stations in the Caribbean Sea. Also, *G. elongatus* basegroups Ia1 and Ia3 have a cosmopolitan distribution whilst basegroup Ia2 was not found in the North Atlantic. The unique basetype detected in *G. ruber ruber* Ia1 was only found in the North Atlantic in our dataset. Unfortunately, the biogeography of the MOTUs of *G. conglobatus* and *G. tenellus* remains unknown due to the low number of observations.

While saturation is also reached at the basegroup level in the global dataset for the three morphospecies, it is not reached for the Indian and North Atlantic oceans, indicating that our sampling was not sufficient in these two basins. Jackknifing analysis indicates that it is likely that two basegroups of *G. ruber* have not been sampled in the North Atlantic, while it is possible that one basegroup of *G. elongatus* may still be discovered in the Pacific Ocean. However, this seems unlikely for *G. elongatus* because the diversity in the Indian Ocean would thus be higher (three observed and four estimated genotypes) than in the global dataset (three observed and estimated genotypes). These results may be the consequence of our unevenly distributed sampling and the fact that the detection of basegroups depends on the fragment of

SSU rDNA covered, which depends of the primer used in each study. Therefore, it is impossible to say whether we failed to capture the diversity in *G. ruber* or *G. elongatus* in every basin, also given the lack of data from the South Atlantic, or if these results reflect an existing bias in our sample set.

We observe a significant difference in the sea surface temperature and mixed layer salinity at which *G. ruber albus* n.subsp., *G. ruber ruber* and *G. elongatus* were collected (Fig 6, Table 2). However, the apparent preference of *G. ruber ruber* for higher salinity may be artificial because most of our sampling for this species originates from the Caribbean and Mediterranean Seas (characterized by higher salinity) and the central Atlantic has not been sampled yet precluding a robust assessment of the true preferences of this taxa. Our sampling suggests differences between the basegroups *G. elongatus* Ia1 and Ia2, which occupy the lower and upper end of the thermal range of the morphological species. We also find *G. ruber albus* n.subsp., *G. ruber albus* n.subsp. Ib and *G. elongatus* Ia2 in more productive waters compared to *G. ruber ruber*, *G. ruber albus* n.subsp. Ia and *G. elongatus* Ia1, but do not observe differences with respect to chlorophyll content or particulate organic carbon. Our dataset does not reveal any seasonality in the occurrence of either the genotypes or basegroups (Fig 4B), but we stress that the sample set may not be suited to reveal such patterns.

## Phylogeny of *Globigerinoides*

The topology and timing of diversification between members of the genus *Globigerinoides* (Figs 2 and 7) is largely congruent with the phylogeny proposed by Aurahs et al. [27]. The deepest split in the molecular clock phylogeny (Fig 7) separates *G. ruber* from *G. conglobatus*, *G. elongatus* and *G. tenellus* and is dated at 17.59 Ma but with a large credible interval on the age of the split (23.25 to 12.58 Ma). The Maximum-likelihood inference (Fig 2) does not support the monophyly of this clade and it is not possible to conclude from the molecular perspective alone if *G. conglobatus* is more closely related to *G. elongatus* and *G. tenellus* or to the *G. ruber* clade. The next diversification event in each lineage occurred in the late Miocene, when *G. conglobatus* diverged from the ancestor of *G. elongatus* and *G. tenellus* (ca. 8.29 Ma) and *G. ruber albus* n.subsp. and *G. ruber ruber* separated (ca. 6.74 Ma). Further diversification occurred between the late Pliocene and early Quaternary, when *G. elongatus* and *G. tenellus* separated concomitantly with the deepest split among the constitutive genotypes of *G. ruber albus* n.subsp. and *G. conglobatus*. A further divergence occurred in the course of the Quaternary between the genotypes Ib and Ic of *G. ruber albus* n.subsp., but all the remaining six divergences at the level of basetypes emerged into the Pleistocene, estimated between ~9 and 224 ka.

## 3D ontogenetic morphology

The largest shell diameter of the analyzed specimens ranges from 250 μm in *G. rubescens* and *G. tenellus*, to 700 μm for *G. conglobatus* (Fig 8), and the CT scans revealed that the specimens consist of 15 to 18 chambers (Fig 9A). The number of chambers is not fixed within a species and specimens with smaller proloculus seem to have more chambers [53]. For example, the chamber number can vary from 15 to 19 chambers in *G. ruber albus*, and the onset of the ontogenetic stage is not tied to the development of a particular chamber [53]. In this study, we use the chamber number as a descriptive term for convenience to explore only our results, and do not mean to imply a fixed boundary between the ontogenetic stages. In all five morphospecies the proloculus is consistently larger than the deuteroconch. Proloculus diameters differ among species, ranging from 9 μm in *G. elongatus* to 17 μm in *G. conglobatus* (Fig 10) and the

ontogenetic development is accompanied by marked differences in the pattern of chamber addition among the species (Figs 9 and 10).

The ontogenetic trajectory of *G. rubescens* is the most stable. It begins with a steady logarithmic increase of chamber size from chambers two to thirteen, then levels off towards chamber 16 and ends with a diminutive final chamber 17 after 3.5 whorls (Figs 9 and 10). While the chamber shape (*S*) remains the same throughout its ontogeny, the whorl expansion rate (*W*) first drops steeply to chamber 5, then decreases slowly until chamber 13, slightly increases until chamber 15 and then decreases again to the final chamber. Inversely, the translation rate (*T*) increases slowly until chamber 13, then drops until chamber 15, to rise sharply over the two last chambers, while the relative distance between the coiling axis and the chamber centroid (*D*) decreases steadily throughout the ontogeny except in chambers 14–15.

Ontogenetic trajectories of *G. tenellus* and *G. elongatus* are initially similar and only diverge in the last stages. The analyzed specimen of *G. tenellus* produced slightly larger chambers but terminated its growth with two chambers less than *G. elongatus* (Fig 10). The chambers of *G. elongatus* gradually flatten between chambers 14–18, resulting in a decreasing S, whilst in *G. tenellus* they become rounder between chambers 14–16, which results in the divergent final shape that distinguishes between the sister species. As for *G. rubescens*, the final chamber of the scanned specimens of *G. tenellus* and *G. elongatus* is smaller than the penultimate chamber, which is indicative of the terminal reproductive stage.

Largest shells are typically found in *G. conglobatus* and *G. ruber*, but shell size is clearly not associated with the growth of more chambers: *G. ruber* has only 15 chambers in our dataset. The ontogenetic trajectory of *G. ruber* differs from all other species in its whorl number, which increases more steeply from chamber 9 onwards (Fig 10) in line with the higher angular increment between successive chambers. However, its expansion rate *W* is close to all other species except for *G. conglobatus* (Fig 10). *G. ruber* and *G. conglobatus* show a higher rate of size increase in consecutive chambers that the other species, such that for *G. ruber* the three last chambers occupy 94% of the total chamber volume (Fig 9B). The rates *D* and *T* are mirrored in their unevenness due to the abrupt decrease of the radius during the ontogeny (see S3 Table), the elevation of the trochospire and the tighter coiling axis. Finally, *G. conglobatus* has the largest test but its most distinctive feature is the increase of the sphericity between chambers 1 to 10 that is followed by compression between chambers 11 to 18. Its whorl expansion rate (*W*) is the highest throughout its ontogeny, but the formation of its high trochospire occurs over the last two chambers with an increase of *T* and a decrease of *D*.

Our Raupian analysis of the 3D ontogenetic trajectory of the five species could be used to determine changes in the position in the growth sequence when the juvenile, near-planispiral, many-chambered stage ends (onset of neanic stage *sensu* Brummer et al. [53]) and when the diagnostic, reproductive morphology is established (onset of adult stage *sensu* Brummer et al. [53]). The distinction of the ontogenetic stages in the CT reconstructions is based mainly on the parameters of chamber addition, but in several cases, the observed transitions could also be correlated with the emergence of further indicative traits, such as supplementary apertures. The analysis of the ontogenetic trajectories reveals that the allocation of chamber number and chamber volume to the ontogenetic trajectory remained similar between *G. rubescens* and *G. ruber* (Fig 9), but the other species show distinct differences in allocation. *G. conglobatus* differs most from the other species, exhibiting distinct juvenile-neanic stage with radially elongated chambers. *G. elongatus* shows a morphologically normal juvenile stage with 10 chambers and becomes trochospiral late in its ontogeny. Both species develop compressed chambers but the compression starts during the neanic stage at chamber 11 for *G. conglobatus* and at the onset of adult stage at chambers 14–15 for *G. elongatus*. By comparison, *G. tenellus* is much smaller, does not develop chamber compression and has fewer chambers (16).

## Discussion

Strict dataset curation of the genetic dataset associated with the application of our nomenclature system confirms recent metabarcoding results which indicate that the biological diversity in planktonic foraminifera is limited [60,61]. We identified only eight genotypes and 14 basegroups within the five sequenced morphospecies of *Globigerinoides*, which likely covers the entire genotypic diversity in the genus. At the basegroup level, *Globigerinoides conglobatus* and *Globigerinoides tenellus* remain undersampled, but for the *Globigerinoides ruber* plexus and *Globigerinoides elongatus*, the sampling effort is sufficient to analyze the distribution of genetic diversity at all hierarchical levels (Fig 5, Table 2).

Our data confirm earlier work [26] in their conclusions that *G. ruber ruber* occurs only in the Atlantic, is the only type with test color and constitutes a single basegroup. We observe no other basegroup or genotype restricted to the Atlantic within the genus (Fig 4), but instead note the apparent absence of the basegroups *G. elongatus* Ia2 as well as *G. ruber albus* n.subsp. Ic1 and potentially Ib2 from the North Atlantic. Despite the fact that our first order Jackknifing (Table 1) and rarefaction analyses (Fig 4) suggest that the diversity in the North Atlantic may not have been captured entirely for *G. ruber albus* n.subsp. at the basegroup level, it does seem to be the case for *G. elongatus* at the basegroup level and for *G. ruber albus* n.subsp. at the genotype level. Therefore, the observed distribution pattern likely highlights an isolation of the tropical Atlantic from the Indian and Pacific Oceans.

Because of the equatorial position of the continents, the subtropical-tropical waters of the world oceans are only connected to a limited degree. At present, transport of tropical/subtropical marine plankton is largely unidirectional, from the Pacific to the Indian Ocean via the Indonesian throughflow, and from the Indian Ocean into the Atlantic via the Agulhas leakage. During glacial times, these connections likely became even more restricted [62]. Indeed, the disappearance of *G. ruber ruber* from the Indian and Pacific Oceans 120 kyrs ago [6] and its persistence in the Atlantic indicate a reduced ability to re-invade the Indian Ocean from the Atlantic. Dispersal from the Indian and Pacific Oceans into the Atlantic via Agulhas leakage is evidenced by the existence of a number of cosmopolitan basetypes (*G. elongatus* Ia1/Ia3 and *G. ruber albus* n.subsp. Ia1/Ia2/Ib2). In this scenario, the absence of *G. elongatus* Ia2 and *G. ruber albus* n.subsp. Ib2/Ic1 in the North Atlantic cannot be the result of dispersal limitation. Instead, the apparent accumulation of recently diverged endemic basegroups in the Pacific rather than the Atlantic (Figs 4 and 5) is reminiscent of the pattern observed in the hyperdiverse *Globigerinella* [63], where it has been ascribed to incumbency (expansion of a species into a new environment being prevented by an incumbent species with similar ecological preferences [64]). In our case, it might be that the Atlantic residents *G. ruber ruber* Ia1, *G. ruber albus* n.subsp. Ia1/Ia2/Ib1 and *G. elongatus* Ia1/Ia3 impede the establishment of invading genotypes recently diverged in the Indian and Pacific Oceans. The lack of diversity in the Atlantic endemic *G. ruber ruber*, compared to the cosmopolitan sister clade (Figs 2 and 4) suggests that no diversification occurs in the North Atlantic. Therefore, the Indian and Pacific Oceans seem to act as the primary source for biodiversity and the North Atlantic as a sink within the *Globigerinoides* genus.

Notwithstanding the pattern of limited connectivity between the Atlantic and the Indian and Pacific Oceans, the majority of the MOTUs has a cosmopolitan distribution within the (sub)tropical habitat of *Globigerinoides*, with co-occurrences at all taxonomic levels at the same stations (Figs 1 and 4), consistent with their apparently similar ecological niches (Fig 6). Although we did not sample *G. ruber ruber* in the South Atlantic, the distribution of the better-covered taxa is associated with higher SST in *G. ruber albus* n.subsp. compared to *G. elongatus* (Fig 6, Table 2). We acknowledge that our sampling of *G. ruber ruber*, with more

sampling stations in the Caribbean and Mediterranean Seas compared to the central Atlantic, may have produced a biased view on the ecological preferences of this morphospecies. However, we are confident that our dataset of *G. ruber albus* and *G. elongatus* does not suffer from this limitation (Fig 4). The difference in thermal niches between *G. ruber albus* n.subsp. and *G. elongatus* has been a matter of debate since the seminal work of Wang [8]. Several studies replicated the observation of the preference of *G. elongatus* for colder waters compared to *G. ruber albus* n.subsp. akin to our observations [9,14–16,18,19,65,66], but observations of the absence of such differences have also been made. Indeed, a global synthesis of seasonally and depth-resolved sediment trap and plankton net observations [11] showed no statistically significant difference between *G. ruber albus* n.subsp. and *G. elongatus* in Mg/Ca composition of the shell. Studies conducted in the Gulf of Mexico [10,21] and in the central North Atlantic [67] showed similar absence of oxygen isotopic offsets between the morphospecies and argued that the difference in habitat, seasonal and calcifying depth is not systematic. Downcore analyses of Mg/Ca ratios from the southwest Pacific [15,20] showed that the difference between the two morphospecies was not stable though time and varied between 0 and 2˚C in temperature space. This is consistent with the findings of Numberger et al. [18] in Mediterranean sediments, who noted oxygen isotopic offsets between the species, but the value and direction of the offset changed during the last 400 kyrs. Altogether, the niches of the two morphospecies may differ, but temperature sensitivity alone is unlikely to be the sole factor explaining the niche difference.

The conflicting observations on the degree of overlap between the ecological niches of *G. ruber albus* n.subsp. and *G. elongatus* raise the question of whether the degree of the overlap could be driven by ongoing diversification at the genotype and basegroup levels. In our analysis, we observe little to no ecological differences between the genotypes and basetypes of *G. ruber albus* n.subsp. and *G. elongatus*, except for (small) differences in temperature, salinity and productivity niches between *G. elongatus* basegroups Ia1 and Ia2 (Fig 6 and Table 2). Therefore, the regionally and temporally varying overlap between the ecological niches of the two morphospecies is unlikely to be the result of ecological differentiation among the constituent MOTUs. There is no evidence for the existence of ecological or biogeographic differentiation between the genotypes of *G. ruber albus* n.subsp. nor *G. elongatus* such as those that were discovered in morphospecies like *Orbulina universa* [68–70], *Globorotalia inflata* [71,72], *Globorotalia truncatulinoides* [73–75], *Globigerina bulloides* [76–79], *Neogloboquadrina pachyderma* [80–83] and *Pulleniatina obliquiloculata* [84,85]. An explanation invoking a vertical niche separation as observed in *Hastigerina pelagica* [86] is unlikely, because *G. ruber albus* n.subsp. and *G. elongatus* are both symbiont-bearing taxa limited to the photic zone and a consistent separation with depth or season would result in a constant isotopic offset, which contrasts general observations (see above).

Although abiotic factors, such as temperature, are important drivers of plankton community structure [87,88], recent studies have shown that biotic interactions may be even more important drivers of plankton diversification. Analyses of plankton metacommunity structure showed that abiotic factors alone explained only 18% of the variability in the distribution of environmental OTUs [89], leaving biotic interactions as the main driver of ecological and biological diversification in the open ocean. Photosymbiosis is the biotic interaction that has been most studied in foraminifera [90] and is of interest to paleoceanographers, not only because it ties photosymbiotic species to photic depths, but also because it impacts the incorporation of stable carbon isotopes and trace elements in the calcareous shell [91–93]. Photophysiology [92,94–98] investigations have documented the dynamic relationship between the foraminifera and their photosymbionts, but the diversity of these interactions, including other interactions such as parasitism or commensalism, has not yet been systematically resolved. Indeed,

Shaked and de Vargas [99] found 21 phylotypes of the dinoflagellate *Symbiodinium* hosted by four morphospecies of tropical planktonic foraminifera, including *G. ruber* and *G. conglobatus*, and suggested that this number most likely represents the lower bound of the true symbiotic diversity, leaving ample space for differentiation due to preference for different symbiont strains.

Planktonic foraminifera, like many protists living in the oligotrophic ocean, are capable of mixotrophy (capable of autotrophy by symbiosis and heterotrophy) and the type of mixotrophy influences the biogeography and seasonality of the mixotrophs hosting the symbionts [100]. We hypothesize that the position in the trophic network occupied by planktonic foraminifera may control when and where they calcify their shell. The control of temperature on planktonic foraminifera individual species abundance and occurrence could be indirect and the physico-chemical condition of the water column that the planktonic foraminifera record may reflect their relationships with other organisms rather than a mere thermal response. In this scenario, temperature alone would not explain evolution in planktonic foraminifera [101] and vital effects impacting the incorporation of carbon isotopes could have varied through time as a function of varying symbiotic association and mixotrophy level [93]. Indeed, a prominent role of biotic factors in the diversification of *Globigerinoides* species is consistent with the lack of physical niche differentiation at the level of genotypes and basegroups. The large number of apparently recently diverging basegroups could result from a high turnover driven by biotic interactions which rarely leads to persistent separation of lineages, resulting in a continuous diversification in the genus throughout the late Neogene and Quaternary (Fig 7), without a clear partitioning of the ecological space along abiotic factors.

Diversification at the cryptic level in the genus likely reflects biotic interactions, but it remains to be explained why and how the morphological evolution and genetic divergence are disconnected at the morphospecies level. For instance, *G. ruber ruber* and *G. ruber albus* n. subsp. diverged around ~6.7 Ma and remained morphologically identical, whereas *G. elongatus* and *G. conglobatus* diverged around 8.3 Ma (Fig 7) but are morphologically distinct from juvenile to adult. Similarly, *G. tenellus* and *G. elongatus*, which are morphologically dissimilar diverged around 2.4 Ma and this event could be concomitant with the divergence time of the constitutive genotype of *G. conglobatus* and *G. ruber albus* n.subsp (Fig 7). Because of a similarity in shape, *G. tenellus* was previously considered a sister species of *G. rubescens*. The apparent similarity motivated us to analyze the ontogeny of this species as well. Our strategy was to recover the potential phylogenetic information contained in the ontogenetic development of the five extant morphospecies of *Globigerinoides* and to use *Globoturborotalita rubescens* as an outgroup. Because of the time-consuming nature of 3D analysis, we limited our approach to a single representative specimen per species to obtain the main differences in the ontogenetic development between species. We acknowledge that intra-species variability in the ontogenetic development exists [53] and that our study design prevents assessing the magnitude of this variability. Nevertheless, the observed contrasting patterns of growth allocation to ontogenetic stages are substantial and associated with systematic changes in chamber shape and growth pattern (Fig 9), in a manner that can be best described in the light of heterochrony [102]. Heterochrony is defined as evolutionary change in the rate and timing of ontogenetic development. Although heterochrony is a concept developed to understand the connection between evolution and development in multicellular organisms, we apply it in a broad sense to planktonic foraminifera because the sequential growth of their tests preserves the sequence of shapes during individual growth. Also, we stress that heterochrony as a concept does not explain the mechanistic cause for evolutionary change, but provides a framework in which the emergence of the divergent adult shapes can be described through changes in the ontogenetic trajectory [102].

In this heterochronic framework, we observe that the *G. rubescens* specimen displays the most stable development with relatively little change in the shape of its chambers during ontogeny compared to the other species (Figs 9 and 10). Considering this morphospecies as outgroup (given its phylogenetic position; Figs 2 and 7), we explore the divergence of adult morphologies of the individual species in terms of Raupian alterations in the ontogenetic trajectory and the successive emergence of new characters. Compared to *G. rubescens*, the morphological innovations in *Globigerinoides* are the emergence of elongate chambers, compressed chambers and supplementary apertures. Chamber elongation is restricted to the juvenile stage of the *G. conglobatus* specimen and it is followed by compression in the neanic-adult stages of large *G. conglobatus*. Chamber compression also occurs in the adult stage of *G. elongatus* and its absence in small *G. tenellus* hints at heterochrony by dwarfing. Supplementary apertures are lacking in the small ancestral *G. rubescens* but are typically found in in the sister clade and their reduction to the last 1–2 chambers in *G. tenellus* is consistent with heterochrony by dwarfing. In *G. tenellus* a single secondary aperture is typically present in the final chamber, whereas all other species of the genus develop at least in the final chambers two supplementary apertures per chamber.

The ratio *S* describing the evolution of the roundness of the chambers is more stable during the ontogeny of *G. ruber* compared to the four other species (Fig 10). The analyzed specimen is large (400 μm) for the few (15) chambers it has, and lacks chamber compression in comparison to *G. elongatus* and *G. conglobatus*, indicating that *G. ruber* may have a neotenic ontogenetic trajectory. Neoteny is characterized by a conservation of juvenile features during the adult stage, reduced compression of the last chamber in the case of *G. ruber*, without a change of size. It is associated with a steeper increase of chamber size at a higher angular increment towards the end of the growth. This scenario would be consistent with the hypothesis that *G. ruber* evolved from *G. obliquus* (which has more compressed chambers) as proposed by Aurahs [27]. In contrast, the ontogenetic trajectory of *G. conglobatus* appears hypermorphic, which is characterized by larger final size. Finally, *G. elongatus* and *G. tenellus* seem to follow similar ontogenetic paths and to differ in the last three chambers, with the compression of the chambers of *G. elongatus* and the increase of the roundness of *G. tenellus* chambers. Also, *G. tenellus* has larger chambers through its ontogeny and its final size is smaller than *G. elongatus*, suggesting progenesis. Progenesis is defined as a loss of an adult feature, the final compressed chamber akin to what we hypothesize for *G. ruber*, but in this case associated with a reduction in size due to a premature interruption of the growth. In terms of size, *G. tenellus* is one of the few known examples of dwarfing in planktonic foraminifera, but unlike the fossil species *Globorotalia exilis*, *Globorotalia miocenica* and *Morozovelloides crassatus* the dwarfing in *G. tenellus* does not (yet) seem to be associated with a reduction of abundance preceding extinction [103].

Evolution through heterochrony could provide an explanation for the erroneous taxonomic placement of *G. elongatus* as a sister to *G. ruber* that led to the informal delimitation *G. ruber* s.l. and s.s. by Wang [8]. Indeed, we hypothesize that *G. elongatus* may not attain the size and shape of *G. conglobatus* because it has smaller chambers, which are less compressed, and could consequently converge towards the size and shape of *G. ruber*. Similarly, *G. tenellus* may create a morphological convergence with *G. rubescens* despite having markedly different pre-adult ontogenetic trajectories (Fig 9). The presence of supplementary apertures in *G. tenellus* is thus an apomorphy of *Globigerinoides*. Based on our observations, we proposed several interpretations of the molecular phylogeny topology that would be in agreement with the morphology, taking into account the heterochronic development within *Globigerinoides* genus (Fig 11).

Similar to previous studies [29,30] our results show that CT-scanning offers a promising avenue for ontogenetic analysis and resolve phylogenetic relationships among extinct species

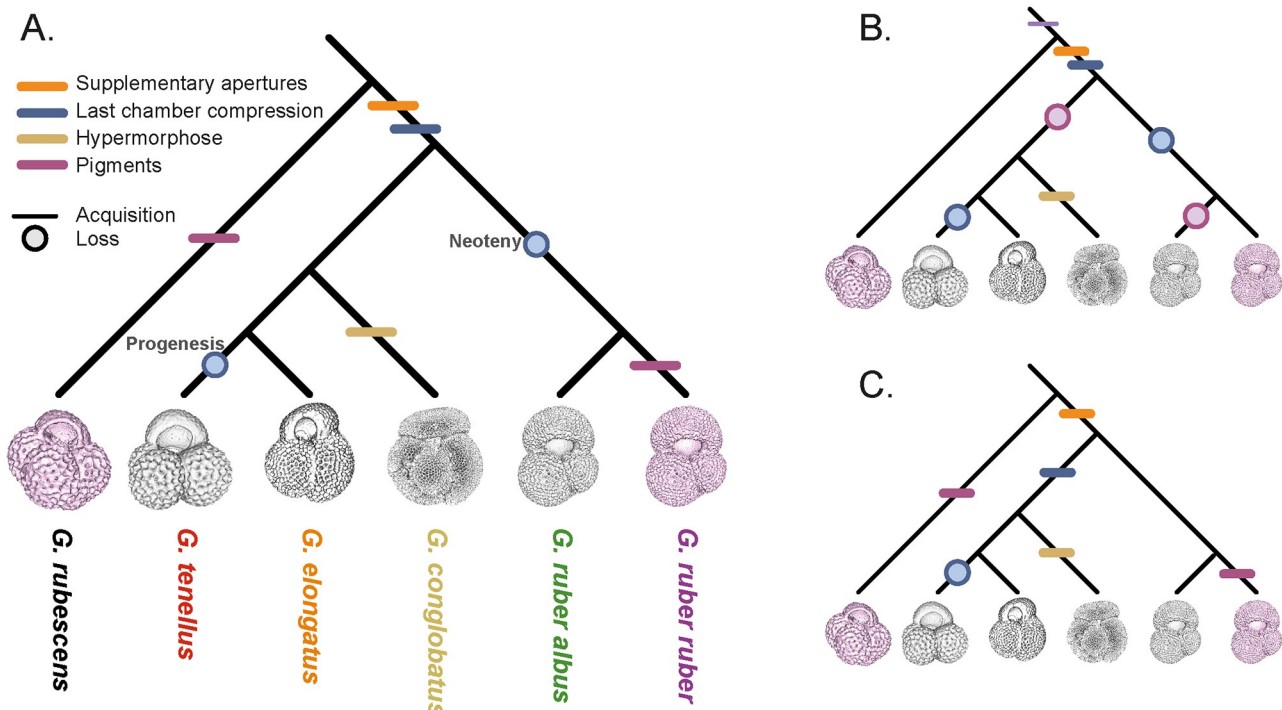

**Fig 11. Cladogram representing the morphological evolution of the Genus *Globigerinoides*.** The cladogram (A) represents the retained scenario and the cladograms (B) and (C) possible but rejected alternatives. (A) The presence of supplementary apertures and compressed last chambers are synapomorphies of the genus. The last compressed chamber is lost in *G. ruber* and *G. tenellus* through neoteny and progenesis respectively. The pink coloration in *G. rubescens* and *G. ruber ruber* is a homoplasic character that appear independently during the evolution of the two species. (B) Alternative scenario where the pink coloration is a synapormophic character of the *Globoturborotalita* and *Globigerinoides* genus but lost in *G. ruber albus* n.subsp. and by the common ancestor of *G. conglobatus*, *G. elongatus* and *G. tenellus*. Although we cannot with certainty choose between the scenario (A) and (B) regarding the pink coloration because the character is not preserved in sediments before 750 ka [6], we prefer the scenario (A) due to its higher parsimony. (C) Alternative scenario where the last compressed chamber is not a synapomorphic character but acquired only in the monophylum *G. conglobatus*, *G. elongatus* and *G. tenellus* and lost by *G. tenellus*. We do not retain this scenario because *Globigerinoides obliquus*, the likely common ancestor of the modern species shows high compression in its last chamber [27].

of planktonic foraminifera [104]. We recognize that we cannot draw firm conclusions from our analysis because of the limited amount of specimen analyzed, and stress the need for replicate analysis to confirm our results. Even though ontogenetic analysis may not explain what triggered the divergence and convergence of juvenile and adult morphologies, it could provide a viable explanation for the apparent disconnection between morphological and genetic divergence. Heterochrony is a process through which large changes in adult morphology could be achieved at genetically low cost [102], creating an impression of large change not matched by the degree of genetic kinship.

## Supporting information

**S1 Fig. Light microscopy images of the specimen CA1261 identified as *Globigerinoides tenellus* and from which sequence match the type IIb of Aurahs et al [26,27].** (a) Umbilical (b) spiral (c) lateral views. The scale bar represents 100 μm.
(TIF)

**S2 Fig. Light microscopy images of the holotype of (C319) and paratypes (C208, C281, C329) of *G. ruber albus* n.subsp.** The archiving museum numbers at the Naturalis

Biodiversity Center, Leiden, The Netherlands are provided below the voucher of the specimens. The scale bar represents 100 μm.
(TIF)

**S1 Table. Metadata and taxonomy of the Sanger sequences used in the study.**
(XLSX)

**S2 Table. Taxonomic equivalence between the existing taxonomic nomenclatures proposed in the literature and our updated molecular taxonomy.**
(XLSX)

**S3 Table. Volume, Cartesian coordinates and parameters of the Raup's model measured on individual chambers of the five selected morphological species (Figs 9 and 10).**
(XLSX)

## Acknowledgments

We thank all crew members and scientist for their help in the collection of planktonic foraminifera. We are thankful to Dr. Yurika Ujiié for her help collecting planktonic foraminifera and for producing genetic data, Dr. Barbara Donner for providing access to the sediment material to produce the CT-scans. We also thank Dr. Julie Meilland for imaging the specimen of *G. tenellus* and the holotype and paratypes of *G. ruber albus* n.subsp. Dr. Willem Renema and Dr. Martina de Freitas Prazeres are acknowledged for their help submitting the holotype and paratypes specimens to the Naturalis Biodiversity Center. We are thankful to Prof. Ralf Schiebel and two anonymous reviewers who provided constructive comments that helped us to improve the present manuscript.

## Author Contributions

**Conceptualization:** Raphaël Morard, Colomban de Vargas, Michal Kucera.

**Data curation:** Raphaël Morard, Mattia Greco, Kate Darling.

**Formal analysis:** Raphaël Morard, Geert-Jan A. Brummer, Mattia Greco, Lukas Jonkers, André Wizemann, Agnes K. M. Weiner, Kate Darling, Michael Siccha, Ronan Ledevin.

**Funding acquisition:** Kate Darling, Hiroshi Kitazato, Thibault de Garidel-Thoron, Colomban de Vargas, Michal Kucera.

**Investigation:** Raphaël Morard, Angelina Füllberg, Geert-Jan A. Brummer, Mattia Greco, Michael Siccha, Michal Kucera.

**Methodology:** Raphaël Morard, Geert-Jan A. Brummer, Michael Siccha.

**Project administration:** Colomban de Vargas, Michal Kucera.

**Resources:** Kate Darling, Hiroshi Kitazato, Michal Kucera.

**Software:** Michael Siccha.

**Supervision:** Raphaël Morard, Michal Kucera.

**Visualization:** Raphaël Morard, Angelina Füllberg.

**Writing – original draft:** Raphaël Morard, Geert-Jan A. Brummer, Mattia Greco, Michal Kucera.

**Writing – review & editing:** Raphaël Morard, Geert-Jan A. Brummer, Mattia Greco, Lukas Jonkers, André Wizemann, Agnes K. M. Weiner, Kate Darling, Michael Siccha, Ronan Ledevin, Thibault de Garidel-Thoron, Michal Kucera.

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
