## [Decision Letter · Decision Letter 0]

1 Aug 2019

PONE-D-19-17658

Genetic and morphological divergence in the warm-water planktonic foraminifera genus Globigerinoides

PLOS ONE

Dear Dr. Morard,

Thank you for submitting your manuscript to PLOS ONE. After careful consideration, we feel that it has merit but does not fully meet PLOS ONE’s publication criteria as it currently stands. Therefore, we invite you to submit a revised version of the manuscript that addresses the points raised during the review process.

I have now received the comments of three external reviewers and as you can see they are mostly positive on the novelty, content and structure of your Ms. Reviewers 2 and 3 are very positive and have only moderate suggestions. On the other hand, reviewer 1 suggests to open a discussion in (paleo-)ecological perspective based on previously published papers and to avoid the proposal of a new naming scheme. The same reviewer also recommends to provide all the available data. Reviewer 2 suggests to clarify the relationships between MOTUs and morphospecies, genotypes and basegroups and also on the potential bias given by the sampling scheme/distribution that might have affected the autoecological interpretations. Reviewer 3 points to the low number of specimens that might not fully support the strong statement in the conclusion.

We would appreciate receiving your revised manuscript by Sep 15 2019 11:59PM. To enhance the reproducibility of your results, we recommend that if applicable you deposit your laboratory protocols in protocols.io, where a protocol can be assigned its own identifier (DOI) such that it can be cited independently in the future. For instructions see: http://journals.plos.org/plosone/s/submission-guidelines#loc-laboratory-protocols

We look forward to receiving your revised manuscript.

Kind regards,

Fabrizio Frontalini

Academic Editor

PLOS ONE

Journal Requirements:

2. In your Methods section, please provide additional location information of the collection sites, including geographic coordinates for the data set if available.

5. We note that Figure 1 in your submission contains a map image which may be copyrighted. All PLOS content is published under the Creative Commons Attribution License (CC BY 4.0), which means that the manuscript, images, and Supporting Information files will be freely available online, and any third party is permitted to access, download, copy, distribute, and use these materials in any way, even commercially, with proper attribution. For these reasons, we cannot publish previously copyrighted maps or satellite images created using proprietary data, such as Google software (Google Maps, Street View, and Earth). For more information, see our copyright guidelines: http://journals.plos.org/plosone/s/licenses-and-copyright.

You may seek permission from the original copyright holder of Figure [1] to publish the content specifically under the CC BY 4.0 license. 

If you are unable to obtain permission from the original copyright holder to publish these figures under the CC BY 4.0 license or if the copyright holder’s requirements are incompatible with the CC BY 4.0 license, please either i) remove the figure or ii) supply a replacement figure that complies with the CC BY 4.0 license. Please check copyright information on all replacement figures and update the figure caption with source information. If applicable, please specify in the figure caption text when a figure is similar but not identical to the original image and is therefore for illustrative purposes only.

Reviewers' comments:

Reviewer's Responses to Questions

**Comments to the Author**

1. Is the manuscript technically sound, and do the data support the conclusions?

Reviewer #1: Partly

Reviewer #2: Yes

Reviewer #3: Partly

2. Has the statistical analysis been performed appropriately and rigorously? 

Reviewer #1: Yes

Reviewer #2: Yes

Reviewer #3: Yes

3. Have the authors made all data underlying the findings in their manuscript fully available?

Reviewer #1: No

Reviewer #2: Yes

Reviewer #3: Yes

4. Is the manuscript presented in an intelligible fashion and written in standard English?

Reviewer #1: Yes

Reviewer #2: Yes

Reviewer #3: Yes

5. Review Comments to the Author

Reviewer #1: The paper of Morard and coauthors on “Genetic and morphological divergence in the warm-water 1 planktonic foraminifera genus Globigerinoides” is a valuable contribution to the understanding of foraminifers. The genus Globigerinoides comprises one of the most abundant group of species in the low latitude ocean, and is ubiquitously used as an archive in paleoceanography. The new data and discussion presented here adds details and thoughts to current state of knowledge, which may improve the use of the Globigerinoides species in paleoceanography and paleoclimate. The data are clearly illustrated in beautiful figures However, the paper may still be improved by opening the discussion by adding references in particular on the (paleo-) ecology of the different species included in the genus, in particular, on G. ruber (e.g., Bijma et al. 1990 and 1992, Jentzen et al. 2018). The seminal paper of Spezzaferri et al. (2015) could be referred to much earlier in the Introduction. Mojtahid et al. (2013) present a great data set and interpretation of G. ruber morphotypes across sapropels over the past 13 ka in the eastern Mediterranean for a better systematic understanding of fossil and modern G. ruber (and other species). The review of Schiebel and Hemleben (2017) presents a comprehensive modern discussion on the ecology and molecular genetics of Globigerinoides. The paper of Burke et al. (2018) add interesting findings on the effect of metabolism on pore size, a discussion, which may be extended to more general thoughts on test morphology.

Whereas the short history of the names, morphotypes, and taxonomy of G. ruber reads nice, I would suggest not to propose a new naming scheme. What we need for a better understanding of the taxonomy and the final use of species in paleoceanography is awareness of the molecular genetics, morphotypes, and ecology. I feel that too much formalisation and too many naming schemes (nicely shown in Fig. 3) rather impede than foster the development of new findings, ideas, and applications of foraminifers in paleoceanography. Many papers have been written on the types of G. ruber, and their presence and absence changes at the regional and basin scale. Finally, each study makes use of the concepts that suit their needs. For example, while philosophising about the morphologies of different types of G. ruber in lines 40-47, shell chemistry (i.e. stable isotopes, Wang 2000) does well indicate different dwelling depths of different morphotypes applicable in paleoclimate at least at the regional scale, i.e. the South China Sea in this case, despite the fact that the morphotypes assigned by Wang may not be formal species. Finally, the taxonomy of the “ruber-group” is not complicated; the confusions with the naming schemes (your nice spaghetti figure) have made it complicated, each of which proposed to make it less confusing (another node to be added to the spaghetti). Not to get me wrong, the present paper is still useful by nicely combining and illustrating the genetic and morphological information; plus improving the ecological information may make this paper a seminal contribution to our field of science (see above).

The International Code of Zoological Nomenclature advises to abstain from the use of subspecies. In the present case of G. ruber, the most simple and elegant solution may be the use of the names G. ruber for the red chromotype and G. albus for the white type.

Line 36 and 605: There are possibly many more morphospecies than five. The genotype G. sacculifer alone includes at least five morphospecies, sac sac, trilobus, quadrilobatus, immaturus, and fistolosus.

Line 46: What do the authors have in mind when evoking „biotic interactions“? Please explain.

Lines 67-68, and following: G. ruber pink did possibly get extinct in the entire Pacific and Indian Oceans, and not only the Indo-Pacific region. Please change wording.

Line 229, and other places: n.subsp. (as well as n.sp. etc.) is not part of the name, and must not be given in italic.

Line 337, and following: Any proof that the color is caused by pigmentation?

Line 371: Very interesting point: Please provide evidence of gametogenic calcification in G. ruber, i.e. SEM images and/or chemical data.

Line 374: The section on “Biogeography and Ecology“ would need to be largely improved, and references to the abundant literature would need to be added (see above). As it stands now, the section rather presents an “Assessment of our G. ruber data base”.

Line 414: see also Aurahs et al. (2009) Bioinformatics and Biology Insights

Lines 475-476: Be careful to not confuse Results and Interpretation: …“indicating that heterochrony may have played a role in the development of the distinct adult morphologies”…

Line 501: change Ocean to Oceans

Line 504: Plankton are transported passively by definition.

Line 504: The transport of tropical plankton may be largely unidirectional toward the west, while the transport in the high latitudes also of the Indian and Pacific Oceans is most toward the east.

Lines 519-520: “…suggests that no diversity is generated…” reads awkward. Please rephrase and make it more to the point.

Line 538: “nearly no differences“ does not read scientific. Either there is a statistically significant difference or not.

Lines 541-543: The paper of Mojtahid et al. (2013) may add data and ideas on the ecology and distribution of G. ruber in the eastern Mediterranean. Please read.

Lines 557-560: “An explanation invoking a vertical niche separation as observed in Hastigerina pelagica [84] is unlikely, because G. ruber albus n.subsp. and G. elongatus are both symbiont-bearing taxa limited to the photic zone and a consistent separation with depth or season would not result in a varying sign of their isotopic offset.” This is possibly not the case. The euphotic zone well includes differences in, e.g., temperature and salinity (i.e. the thermocline) and varying light levels, and stratification of water bodies and species niches is the normal case. This includes effects on isotopes and element ratios.

Line 567: Please refer to the papers of Takagi and co-authors, including the new paper under discussion: Biogeosciences Discuss., https://doi.org/10.5194/bg-2019-145

Line 577: change oceans to ocean

Lines 594-ff: The following paragraph read rambling. Stay to the point, and avoid general statements that cannot be proven. For example, please refer to your figures to prove the statements given in Lines 597-599; the same in lines 599-601.

Line 602: considered

Line 635: “The chambers of G. ruber remain spherical throughout its ontogeny...“; this is not the case. Please see Plate 2.8 in Schiebel and Hemleben (2017). Chambers in juvenile G. ruber are compressed. This can also be seen in your own Fig. 9. This affects the statements in the following paragraphs, which may need some rethinking.

Fig. 5: Making all y-axis the same length, i.e. 0 – 16, may improve immediate understanding of the message presented here.

Line 1036: change to “Fig 3. Development and consistency across the nomenclatural scheme...“

Lines 1098 and 1100: represents

Please revise references, e.g.:

11 and 25: delete Elsevier B.V

47 and 53: delete “Available”

92 is incomplete

I would assume that all data will be made available.

Reviewer #2: This paper provides a thorough overview of the genetic and morphological evolution of the extant species within the genus Globigerinoides. Globigerinoides is one of the most abundant modern genera, but relationships within this clade are still debated. Multiple different methods are used to investigate how the species within this genus are related, to investigate their ecological preferences, and to hypothesise how observed differences in morphology could have developed. They suggest the recognition of subspecies to describe the two colour morphs of G. ruber, and use genetic evidence to demonstrate that Globoturborotalita tenella should be considered a member of this genus.

I found the comprehensive nature of this study of Globigerinoides gives an important overview of the taxonomy that is likely to be useful to all people working with recent planktonic foraminifera. The broad range of methods used provide clear support for the hypotheses developed in this study. Generally, I find it a very well written document, although there are a few points that could be improved.

It would be helpful to make the relationship between the MOTUs and the basegroups / genotypes / morphotypes clearer. For example, the caption / text for Figure 2 refers to the MOTUs, but the figure shows morphospecies / genotypes / basegroups. Similarly the text for Figure 5 (l188) refers to MOTUs lvl-2 / lvl-3, whereas the figure refers to genotypes / basegroups.

In discussion of the influence of environment on morphotype genotype distributions (Fig. 6, Table 2, L402-410) it is suggested that G. ruber ruber has a preference for cooler and more saline environmental conditions than G. ruber albus. However, this may be due to the sampling biases associated with the sampling distribution. G. ruber ruber is limited to the Atlantic ocean, where it is found in the vast majority of samples, suggesting that the Atlantic data points represent a more limited environmental range than the other oceans.

Table 1 suggests that the So falls outside the 95% confidence interval from Se for multiple of the measurements, e.g. for North Atlantic G. ruber and G. elongatus Basegroup. However the numbers in the table seem to disagree: 7.97 – 2.71 < 6, implying So falls within the CI95.

Minor points

• L45/46. This should read ‘…either…or…” rather than “…neither…nor…”

• L122. “The specimen” should be plural, i.e. “The specimens”

• I’m assuming the NCBI accession numbers and the museum number will be filled in before the manuscript is published, e.g. l136

• L172, for clarity, it would be helpful to change “… proposed by ABGD and PTP…” to “… proposed by either ABGD or PTP…”.

• “n. subsp.” should not be italicised throughout, e.g. l229, l281.

• L406, “G. ruber albus” should be italicised

• L548-549 “Only a small difference in preferred habitat temperature between G. ruber albus n. subsp. genotypes Ia and Ic is observed”. Table 2 / Fig 6 seem to show no difference between these two genotypes.

• L550. Should refer to Fig. 6 not Fig. 9

• L635. I think this should refer to Fig. 10 not Fig. 6

• L776, Reference 29 should be “Frontiers”

• Tables 1 / 2 should use “.” not “,” to indicate decimal places.

• Figure 5 needs more detailed labels or a more detailed caption – it’s currently not clear what the four different plots are showing. Putting the titles outside the figures would make this clearer.

• Fig. 6 caption, l1051, the second ‘ruber’ should be italicised.

• Fig. 11. What is the green line on the G. ruber branch of cladogram C?

Reviewer #3: I enjoyed reading the manuscript by Morard et al, which describes both genetic and morphological evidence for divergence in the Globigerinoides genus. Species in this genus are frequently used in paleoceanographic studies, and therefore understanding inter- and intraspecific variation in genetics, ecology and morphology is crucial to interpret past environmental reconstructions. The authors provide a comprehensive overview of both genetic and ecological data on all extant species of the genus as well as a new addition, and also include a first assessment on diverging ontogenetic trajectories among species.

My only issue with the study concerns the number of specimens used for morphological analyses. Initially the authors point out that analysing one specimen per species will only provide a rough first assessment of ontogenetic trajectories among species, but in the last four paragraphs many stronger conclusions are drawn from these single-specimen analyses. Although intraspecific ontogenetic variation is likely smaller than among-species trajectories, differences within species still likely exist. For example, do all specimens within a species build the exact same number of chambers? If not, which ontogenetic phases could have varying chamber numbers and how would that affect the overall trajectories? All analysed specimens here have 15-18 chambers, so even the addition of one extra chamber changes the results. Additionally, especially the adult stage of many species is known to possess a great degree of intraspecific morphological variation. To test whether the adult stages of for example G. tenellus and G. elongatus are statistically different, or just end-members of a larger overlapping cloud in morphospace more specimens are needed. CT-scanning is an expensive and time-consuming task which reduces the number of specimens to feasibly analyse, but even another handful of specimens per species would greatly help to determine differences between inter- and intra-specific ontogenetic trajectories among species.

Minor comments

Line 52-53: Of >100 Neogene biostratigraphic events described by Wade et al (2011), only 8 are from Globigerinoides species and none are zonal markers. To say that the genus represents a cornerstone for biostratigraphy seems exaggerated.

Line 63, 69: Add full stops in G pyramidalis (2x)

Line 77: s.s.

Line 115: briefly explain in the Methods why Globoturborotalita rubescens was included in this study on the Globigerinoides genus. Is there any pre-existing genetic/morphological evidence that it might be better placed in Globigerinoides?

Line 136: will accession numbers be added in the final manuscript?

Line 238: do all specimens of a given species have the same number of chambers? If not, large size might just be due to a higher number of chambers. How would ontogenetic trajectories change for different numbers of chambers? Which life stages would appear longer or shorter? Unless all specimens build exactly the same number of chambers there will likely be some ontogenetic variation within species, so a larger number of specimens per species needs to be analysed to support claims regarding the decoupling between genetic and morphological diversification.

Line 265, 267: why is the number of sequences mentioned in these lines different? Please explain/adjust.

Lines 512-518: with roughly 1 foram per litre of sea water, competition among foraminifera specimens is likely weak. Is this weak competition enough to explain the presence/absence of specific genotypes in different ocean basins?

Line 602: change 'consider' to 'considered'.

Lines 644-646: even if ontogenetic trajectories are similar within species, the final adult forms contain a lot of intraspecific variability. To check whether the last three chambers of G. tenellus and G. elongatus are statistically different in morphospace more specimens need to be analysed.

6. PLOS authors have the option to publish the peer review history of their article (what does this mean?). If published, this will include your full peer review and any attached files.

Reviewer #1: Yes: Ralf Schiebel

Reviewer #2: No

Reviewer #3: No

---

## [Author Response · Author response to Decision Letter 0]

27 Sep 2019

In the following, the comments of the reviewers are indicated by these symbols ***…*** and our responses are indicated by these symbols >>> …. <<<. We specify when necessary the page number of the changes we have made in the modified version of the manuscript with track changes at the end of our answers. 

***Reviewer #1: The paper of Morard and coauthors on “Genetic and morphological divergence in the warm-water 1 planktonic foraminifera genus Globigerinoides” is a valuable contribution to the understanding of foraminifers. The genus Globigerinoides comprises one of the most abundant group of species in the low latitude ocean, and is ubiquitously used as an archive in paleoceanography. The new data and discussion presented here adds details and thoughts to current state of knowledge, which may improve the use of the Globigerinoides species in paleoceanography and paleoclimate. The data are clearly illustrated in beautiful figures However, the paper may still be improved by opening the discussion by adding references in particular on the (paleo-) ecology of the different species included in the genus, in particular, on G. ruber (e.g., Bijma et al. 1990 and 1992, Jentzen et al. 2018). The seminal paper of Spezzaferri et al. (2015) could be referred to much earlier in the Introduction. Mojtahid et al. (2013) present a great data set and interpretation of G. ruber morphotypes across sapropels over the past 13 ka in the eastern Mediterranean for a better systematic understanding of fossil and modern G. ruber (and other species). The review of Schiebel and Hemleben (2017) presents a comprehensive modern discussion on the ecology and molecular genetics of Globigerinoides. The paper of Burke et al. (2018) add interesting findings on the effect of metabolism on pore size, a discussion, which may be extended to more general thoughts on test morphology. ***

>>>We are grateful to the referee for acknowledging the potential of our results. Whilst we agree on the merit of discussing the paleoecology of the constituent morphospecies for the interpretation of proxies, we fear that this would detract from the main message of the manuscript and can only partly be addressed by our data. The main reason is that morphospecies consist of multiple genetic types, whose ecology and biology (especially, as we show) likely differs, but whose identity cannot be determined from fossil material. Even the often cited morphotypes of G. ruber in fact only allow recognising G. ruber albus from G. elongatus. Since the main focus of our study was to understand the process of diversification in the tropical foraminifera at the genetic level, we propose to defer the discussion on the paleoecology of the constituent morphospecies for another study. Nevertheless we are citing some of the suggested literature by the reviewer in the introduction when it is suited.<<<

***Whereas the short history of the names, morphotypes, and taxonomy of G. ruber reads nice, I would suggest not to propose a new naming scheme. What we need for a better understanding of the taxonomy and the final use of species in paleoceanography is awareness of the molecular genetics, morphotypes, and ecology. I feel that too much formalisation and too many naming schemes (nicely shown in Fig. 3) rather impede than foster the development of new findings, ideas, and applications of foraminifers in paleoceanography. Many papers have been written on the types of G. ruber, and their presence and absence changes at the regional and basin scale. Finally, each study makes use of the concepts that suit their needs. For example, while philosophising about the morphologies of different types of G. ruber in lines 40-47, shell chemistry (i.e. stable isotopes, Wang 2000) does well indicate different dwelling depths of different morphotypes applicable in paleoclimate at least at the regional scale, i.e. the South China Sea in this case, despite the fact that the morphotypes assigned by Wang may not be formal species. Finally, the taxonomy of the “ruber-group” is not complicated; the confusions with the naming schemes (your nice spaghetti figure) have made it complicated, each of which proposed to make it less confusing (another node to be added to the spaghetti).***

>>> We understand the point made by the reviewer and we agree that producing new naming schemes for every publication may seem bewildering and impeding efficient scientific communication. However, we cannot help the fact that previous studies have lead to the development of multiple, partly incompatible naming schemes. Such development is inevitable for every “age of exploration” and it is also the reason why biological nomenclature had to be codified and formalised. We have now entered a new stage, where, instead of exploration, we can carry out global syntheses. This allows us, unlike the previous efforts, to devise a consistent and globally applicable and stable nomenclature. In anticipation of this development, we have made an effort earlier in designing a workflow for such molecular nomenclature in a separate publication (Morard et al., 2016). We have applied this method for this time on the microperforate clade (Morard et al., 2019) and find it imperative to use it to consolidate the molecular nomenclature of the studied group as well. We strongly advise against the perpetuation of the concept of various morphotypes within G. ruber, unless their biological nature has been determined. We note that the morphotypes the reviewer is referring to are largely reflecting the division between G. ruber and G. elongatus. This has been shown already by Aurahs et al. (2011) and has nothing to do with the designation of names for genetic types.

One could, of course, rightfully ask, if a naming scheme is necessary at all. We believe it is and we laid out the reasons clearly in Morard et al. (2016). If we had no names for genetic types, there would be no possibility to refer to their existence, distribution or ecology. If used an arbitrary scheme, we would only contribute to proliferation of names and confusion. The objective nomenclatural scheme we propose will instead help to achieve the long-term scientific goal of connecting biologically meaningful entities with names and with data on their occurrence and ecology.

<<<

***Not to get me wrong, the present paper is still useful by nicely combining and illustrating the genetic and morphological information; plus improving the ecological information may make this paper a seminal contribution to our field of science (see above).

The International Code of Zoological Nomenclature advises to abstain from the use of subspecies. In the present case of G. ruber, the most simple and elegant solution may be the use of the names G. ruber for the red chromotype and G. albus for the white type. ***

>>> We agree that naming the two variants simply G. ruber and G. albus instead of G. ruber ruber and G. ruber albus would be easier and more elegant but unfortunately, the colour is the only morphological feature that distinguish them, and it is not preserved beyond 750 kyrs (indicated in lines 313 to 322). Thus, we are confronted with a case where the diagnostic character cannot be used throughout the range (in this case stratigraphic range) of the species. In such case, the ICZN recommends to designate at a subspecies level. We are not aware of any recommendation of the ICZN that would speak against the use of subspecies. On the contrary, the Code contains rich and clear instructions indicating that subspecies names are its integral part (Article 5.2, Article 11.4.2)).<<<

***Line 36 and 605: There are possibly many more morphospecies than five. The genotype G. sacculifer alone includes at least five morphospecies, sac sac, trilobus, quadrilobatus, immaturus, and fistolosus. ***

>>> In both places, we will clarify that we only refer to the extant members of the genus. There are of course many more fossil members of the genus, but for the extant ones, we are not aware of any other commonly recognised morphospecies in Globigerinoides. The fact that Trilobatus sacculifer consists of multiple morphospcies (of which at least two have been widely recognised) is an exception as highlighted by André et al. (2013). Lines 36 and 637.<<<

***Line 46: What do the authors have in mind when evoking „biotic interactions“? Please explain. ***

>>>We provide symbiosis as an example that is the most obvious type of biotic interaction that we have in mind. We provide more detailed explanation in the discussion section. Line 47. <<<

***Lines 67-68, and following: G. ruber pink did possibly get extinct in the entire Pacific and Indian Oceans, and not only the Indo-Pacific region. Please change wording. ***

>>>Modification made. Line 68. <<<

***Line 229, and other places: n.subsp. (as well as n.sp. etc.) is not part of the name, and must not be given in italic. ***

>>> We have made the correction throughout the document.<<<

***Line 337, and following: Any proof that the color is caused by pigmentation? ***

>>> Not to our knowledge. We have replaced pigmentation by “color” to be more cautious throughout the document L-340. <<<

***Line 371: Very interesting point: Please provide evidence of gametogenic calcification in G. ruber, i.e. SEM images and/or chemical data. ***

>>>The referee is right that the presence of a gametogenic calcification in G. ruber is poorly constrained and a careful search of the literature revealed that most cases where it has been invoked refer to a comparison with T. sacculifer. We therefore deleted this information from the description of the terminal stage. Lines 380-381<<<

***Line 374: The section on “Biogeography and Ecology“ would need to be largely improved, and references to the abundant literature would need to be added (see above). As it stands now, the section rather presents an “Assessment of our G. ruber data base”. ***

>>> In the spirit of our response to the initial comment by the referee we have renamed the section into “Distribution and ecological preferences of Globigerinoides MOTUs” to emphasise that we are here presenting the distribution and ecology of genetic types. Since we include all sequence data from the literature, the description of our database is entirely comprehensive of all studies that have ever generated genetic data from this species. Line 384<<<

***Line 414: see also Aurahs et al. (2009) Bioinformatics and Biology Insights***

>>>We here cite the work of Aurahs et al. (2011) whose philosophy is entirely based on the work presented in Aurahs et al. (2009) but with a more extensive dataset. We therefore consider that citing only Aurahs et al. (2011) is sufficient.<<<

***Lines 475-476: Be careful to not confuse Results and Interpretation: …“indicating that heterochrony may have played a role in the development of the distinct adult morphologies”… ***

>>>We have removed this sentence which was indeed not at its place in the Result section. Lines 496-497.<<<

***Line 501: change Ocean to Oceans***

>>>Corrected. Line 522.<<<

***Line 504: Plankton are transported passively by definition. ***

>>> We have removed the “passive” from the sentence to avoid semantic redundancy. Line 524. <<<

***Line 504: The transport of tropical plankton may be largely unidirectional toward the west, while the transport in the high latitudes also of the Indian and Pacific Oceans is most toward the east. ***

>>> We have added that the westwards transport occur for tropical marine but we prefer not add the part on the eastwards transport of high latitude plankton to not lose focus. Line 525.<<<

***Lines 519-520: “…suggests that no diversity is generated…” reads awkward. Please rephrase and make it more to the point. ***

>>>We have rephrased the sentence. Line 541.<<<

***Line 538: “nearly no differences“ does not read scientific. Either there is a statistically significant difference or not. ***

>>> We have removed “Nearly”. Line 565.<<<

***Lines 541-543: The paper of Mojtahid et al. (2013) may add data and ideas on the ecology and distribution of G. ruber in the eastern Mediterranean. Please read. ***

>>>We have considered the work of Mojtahid et al. (2013) to be included in the section of the article but we believe that the reviewer was actually mentioning Mojtahid et al. (2015) “Thirteen thousand years of southeastern Mediterranean climate variability inferred from an integrative planktic foraminiferal‐based approach”. In this part of the discussion, we are discussing the ecological differentiation between G. ruber and G. elongatus, in particular regarding their relationship to temperature. We discuss the fact that the ecological separation between G. ruber and G. elongatus is not systematic and we based our argumentation on isotopic studies. The study of Mojtahid et al. (2015) reports counts of G. ruber and G. elongatus together with size measurements. Although they make compelling observation in size variation through time we did not find information that would help discuss the potential niche partitioning between the species because no isotopic nor trace elements measurements are presented in this paper. However, this manuscript is cited in the introduction instead.<<<

***Lines 557-560: “An explanation invoking a vertical niche separation as observed in Hastigerina pelagica [84] is unlikely, because G. ruber albus n.subsp. and G. elongatus are both symbiont-bearing taxa limited to the photic zone and a consistent separation with depth or season would not result in a varying sign of their isotopic offset.” This is possibly not the case. The euphotic zone well includes differences in, e.g., temperature and salinity (i.e. the thermocline) and varying light levels, and stratification of water bodies and species niches is the normal case. This includes effects on isotopes and element ratios. ***

>>>We have reformulated this part of the discussion. We cite several works that show that when there is an isotopic offset between the species, which is not always the case, the value and the sign of the offset is not constant. It means that when there is an offset, G. elongatus can either display preferences to colder or warmer temperature compared to G. ruber albus, which is not possible if there is a constant depth and/or niche separation between the two species. Lines 588-589.<<<

***Line 567: Please refer to the papers of Takagi and co-authors, including the new paper under discussion: Biogeosciences Discuss., https://doi.org/10.5194/bg-2019-145***

>>>We cite already four papers of Takagi and co-authors. We added the paper which is currently under discussion at Biogeosciences. Line 598.<<<

Line 577: change oceans to ocean

>>>Corrected. Line 606.<<<

Lines 594-ff: The following paragraph read rambling. Stay to the point, and avoid general statements that cannot be proven. For example, please refer to your figures to prove the statements given in Lines 597-599; the same in lines 599-601.

>>>We have reformulated part of the paragraph to be closer to our observation and also refer to our figure when necessary. Lines 623-6635.<<<

Line 602: considered

>>>Corrected. Line 635.<<<

Line 635: “The chambers of G. ruber remain spherical throughout its ontogeny...“; this is not the case. Please see Plate 2.8 in Schiebel and Hemleben (2017). Chambers in juvenile G. ruber are compressed. This can also be seen in your own Fig. 9. This affects the statements in the following paragraphs, which may need some rethinking.

>>>We have corrected the statement by saying that the roundness of the chambers of G. ruber remain stable throughout its ontogeny. We have amended the rest of the paragraph accordingly. Line 668-670. <<<

Fig. 5: Making all y-axis the same length, i.e. 0 – 16, may improve immediate understanding of the message presented here.

>>>We have modified the figure following the reviewer’s recommendation.<<<

Line 1036: change to “Fig 3. Development and consistency across the nomenclatural scheme...“

>>>Corrected. Line 1090.<<<

Lines 1098 and 1100: represents

>>> Corrected. Lines 1155 and 1159.<<<

Please revise references, e.g.:

11 and 25: delete Elsevier B.V

47 and 53: delete “Available”

92 is incomplete

>>> The reference list has been corrected.<<<

I would assume that all data will be made available.

>>>The reviewer assumes correctly, the data have been deposited on NCBI under the accession number MN383323 to MN384218. It is common practice in molecular biology to deposit the data after the first assessment of the manuscript by reviewers.<<<

Reviewer #2: This paper provides a thorough overview of the genetic and morphological evolution of the extant species within the genus Globigerinoides. Globigerinoides is one of the most abundant modern genera, but relationships within this clade are still debated. Multiple different methods are used to investigate how the species within this genus are related, to investigate their ecological preferences, and to hypothesise how observed differences in morphology could have developed. They suggest the recognition of subspecies to describe the two colour morphs of G. ruber, and use genetic evidence to demonstrate that Globoturborotalita tenella should be considered a member of this genus.

I found the comprehensive nature of this study of Globigerinoides gives an important overview of the taxonomy that is likely to be useful to all people working with recent planktonic foraminifera. The broad range of methods used provide clear support for the hypotheses developed in this study. Generally, I find it a very well written document, although there are a few points that could be improved.

It would be helpful to make the relationship between the MOTUs and the basegroups / genotypes / morphotypes clearer. For example, the caption / text for Figure 2 refers to the MOTUs, but the figure shows morphospecies / genotypes / basegroups. Similarly the text for Figure 5 (l188) refers to MOTUs lvl-2 / lvl-3, whereas the figure refers to genotypes / basegroups.

>>> We have made the relationship clearer. As indicated in the method section in Lines 148 181, MOTUs lvl-2 are equivalent to genotypes and MOTUs lvl-3 are equivalent to basegroup but we agree that this should be clear throughout the text. We have modified the Figures 2 and 5 accordingly and also modified the caption of the figures. Lines 1095-1104.<<<

In discussion of the influence of environment on morphotype genotype distributions (Fig. 6, Table 2, L402-410) it is suggested that G. ruber ruber has a preference for cooler and more saline environmental conditions than G. ruber albus. However, this may be due to the sampling biases associated with the sampling distribution. G. ruber ruber is limited to the Atlantic ocean, where it is found in the vast majority of samples, suggesting that the Atlantic data points represent a more limited environmental range than the other oceans.

>>> The reviewer is right that our sampling is probably inducing a bias in the analysis. We have stressed that point in the result and the method sections (Lines 415-419 and Lines 549-553)<<<

Table 1 suggests that the So falls outside the 95% confidence interval from Se for multiple of the measurements, e.g. for North Atlantic G. ruber and G. elongatus Basegroup. However the numbers in the table seem to disagree: 7.97 – 2.71 < 6, implying So falls within the CI95.

>>>We agree that the Jackniffing results may be partly compromised by sampling bias and we indicate this point in the result section in the lines 405-412. However, we confirm that the So falls outside of the 95% confidence interval in the case mentioned by the reviewer: 7.97 – (2.71/2) = 6,615 > Se. The lower and upper limits of the CI95 have to be calculated as Se± (CI95/2), not Se ± (CI95).<<<

Minor points

• L45/46. This should read ‘…either…or…” rather than “…neither…nor…”

>>>Corrected. Line 46.<<<

• L122. “The specimen” should be plural, i.e. “The specimens”

>>> Corrected. Line 124.<<<

• I’m assuming the NCBI accession numbers and the museum number will be filled in before the manuscript is published, e.g. l136

>>>Yes. Line 138.<<<

• L172, for clarity, it would be helpful to change “… proposed by ABGD and PTP…” to “… proposed by either ABGD or PTP…”.

>>>Corrected. Line 176.<<<

• “n. subsp.” should not be italicised throughout, e.g. l229, l281.

>>> We have corrected all the occurrences.<<<

• L406, “G. ruber albus” should be italicised

>>>Corrected. Line 421.<<<

• L548-549 “Only a small difference in preferred habitat temperature between G. ruber albus n. subsp. genotypes Ia and Ic is observed”. Table 2 / Fig 6 seem to show no difference between these two genotypes.

>>>That is correct, we have removed this part. Lines 575-576.<<<

• L550. Should refer to Fig. 6 not Fig. 9

>>>Corrected. Line 578.<<<

• L635. I think this should refer to Fig. 10 not Fig. 6

>>>Corrected. Line 669.<<<

• L776, Reference 29 should be “Frontiers”

>>>Corrected.<<<

• Tables 1 / 2 should use “.” not “,” to indicate decimal places.

>>>We have made the correction. Lines 1171-1182.<<<

• Figure 5 needs more detailed labels or a more detailed caption – it’s currently not clear what the four different plots are showing. Putting the titles outside the figures would make this clearer.

>>>We have modified the figure accordingly to the reviewer recommendation. We have also completed the figure caption. Lines 1102-1104.<<<

• Fig. 6 caption, l1051, the second ‘ruber’ should be italicised.

>>>Corrected. Line 1108.<<<

• Fig. 11. What is the green line on the G. ruber branch of cladogram C?

>>>It was a mistake which has been removed. We have modified the Figure 11 to ease its reading.<<<

Reviewer #3: I enjoyed reading the manuscript by Morard et al, which describes both genetic and morphological evidence for divergence in the Globigerinoides genus. Species in this genus are frequently used in paleoceanographic studies, and therefore understanding inter- and intraspecific variation in genetics, ecology and morphology is crucial to interpret past environmental reconstructions. The authors provide a comprehensive overview of both genetic and ecological data on all extant species of the genus as well as a new addition, and also include a first assessment on diverging ontogenetic trajectories among species.

My only issue with the study concerns the number of specimens used for morphological analyses. Initially the authors point out that analysing one specimen per species will only provide a rough first assessment of ontogenetic trajectories among species, but in the last four paragraphs many stronger conclusions are drawn from these single-specimen analyses. Although intraspecific ontogenetic variation is likely smaller than among-species trajectories, differences within species still likely exist. For example, do all specimens within a species build the exact same number of chambers? If not, which ontogenetic phases could have varying chamber numbers and how would that affect the overall trajectories? All analysed specimens here have 15-18 chambers, so even the addition of one extra chamber changes the results. 

>>>The reviewer is right to point out the limitation of morphological data being available from only one specimen per species. At present, the data acquisition (interpretation of CT scans) has to be carried out manually and can require several weeks per specimen (Especially for G. conglobatus). However, the information provided by this analysis is extremely informative and very likely bears an enormous potential for the understanding of the evolution of planktonic foraminifera. Perhaps a fair way to look at the data we have is that they provide an additional clue on how to reconcile the incongruence between morphological and molecular phylogenies.

The referee is also right that referring to stage transitions by chamber number would only make sense if the number is constant within the species. In our case, we use it only as a descriptive term, which is only possible because we have one specimen per species. In reality, the number of chamber is indeed variable within each species as observed by Brummer et al. (1987), and can vary between 15 and 19 per specimen for G. ruber. Brummer et al. (1987) also showed that the onset of ontogenetic stages was variable between individuals of the same species, and the transition from one stage to another is not tied to a chamber number. We have clarified these points in the result section (Lines 446-451) and we have also toned down the conclusion drawn in the last four paragraphs (Lines 668-726). <<<

Additionally, especially the adult stage of many species is known to possess a great degree of intraspecific morphological variation. To test whether the adult stages of for example G. tenellus and G. elongatus are statistically different, or just end-members of a larger overlapping cloud in morphospace more specimens are needed. CT-scanning is an expensive and time-consuming task which reduces the number of specimens to feasibly analyse, but even another handful of specimens per species would greatly help to determine differences between inter- and intra-specific ontogenetic trajectories among species.

>>> We could not agree more with the reviewer. Replication would allow in the future to constrain the variability in ontogenetic trajectories and unravel more mechanistically at what points of growth do stage transitions occur. Because of the lack of replication, we have tried to be careful when framing the discussion and we thank the referee for point out some places (see below) where the statements could be hedged even better. For the present manuscript, we believe the existing CT-scan dataset is sufficient to elucidate from an independent angle the contradictory results between morphological and molecular phylogenies in Globigerinoides. Although we do not provide a strongly supported answer on the exact phylogenetic process, we hope that our study will motivate future work to create a framework in which CT-scans will be used to implement the ontogeny into phylogenetic reconstruction of Globigerinoides and foraminifera. Doing so with the appropriate amount of replication will necessitate substantial resources and our preliminary results may help justifying future work in this direction.<<<

Minor comments

Line 52-53: Of >100 Neogene biostratigraphic events described by Wade et al (2011), only 8 are from Globigerinoides species and none are zonal markers. To say that the genus represents a cornerstone for biostratigraphy seems exaggerated.

>>>We have removed this part of the sentence. Lines 53<<<

Line 63, 69: Add full stops in G pyramidalis (2x)

>>>Corrected. Lines 63 and 70.<<<

Line 77: s.s.

>>>Corrected. Line 78.<<<

Line 115: briefly explain in the Methods why Globoturborotalita rubescens was included in this study on the Globigerinoides genus. Is there any pre-existing genetic/morphological evidence that it might be better placed in Globigerinoides?

>>> We have indicated that G. rubescens was included in the analysis to serve as an outgroup for phylogenetic analysis. Lines 119-120.<<<

Line 136: will accession numbers be added in the final manuscript?

>>>Yes, it is a common practice to add the accession number after the first assessment of the manuscript. Line 138.<<<

Line 238: do all specimens of a given species have the same number of chambers? If not, large size might just be due to a higher number of chambers. How would ontogenetic trajectories change for different numbers of chambers? Which life stages would appear longer or shorter? Unless all specimens build exactly the same number of chambers there will likely be some ontogenetic variation within species, so a larger number of specimens per species needs to be analysed to support claims regarding the decoupling between genetic and morphological diversification.

>>>As discussed above, the number of chamber per specimen is variable. Brummer et al. (1987) showed that the number of chamber in G. ruber albus varied between 15 and 19 chambers in the specimen analysed. Brummer et al. (1987) also suggested that the onset of stages was depending of the absolute size of the specimen and not the number of chambers, and that in general, specimens with smaller proloculus added more chambers during their ontogeny. We reiterate here that our argumentation on the heterochrony is based on the occurrence or absence of adult features in the ontogenetic sequence, namely the compression of the last chambers, and not the absolute number of chambers, but we agree that the delineation of the ontogenetic stages cannot be carried out strictly by reference to chamber number. We chose this approach because we only analysed one specimen per species, and it that situation it was easier to compare the life stages with the number of chambers in the text, but we also refer to the position of the stage transitions in terms of the proportion of growth – chamber allocation (given in Fig. 9B). <<<

Line 265, 267: why is the number of sequences mentioned in these lines different? Please explain/adjust.

>>>After a careful check, we realized that we used this number inconsistently here and we have adjusted the number. We thank the reviewer to have noticed it. Lines 271, 273.<<<

Lines 512-518: with roughly 1 foram per litre of sea water, competition among foraminifera specimens is likely weak. Is this weak competition enough to explain the presence/absence of specific genotypes in different ocean basins?

>>>The competition in plankton may not work as a direct antagonist interaction but rather as a relative success to gather resources to complete life cycle and achieve successful reproduction. The concept of incumbency which is mentioned here argues that it is difficult for a new population (basegroup in that case) to be established and grow while a population with similar trophic strategy is already established. Lines 536-537.<<<

Line 602: change 'consider' to 'considered'.

>>>Corrected. Line 635.<<<

Lines 644-646: even if ontogenetic trajectories are similar within species, the final adult forms contain a lot of intraspecific variability. To check whether the last three chambers of G. tenellus and G. elongatus are statistically different in morphospace more specimens need to be analysed.

>>>Again, we agree with the reviewer. We have reformulated the sentence to be more nuanced in the absence of rigorous statistical test. . Lines 679.<<<

---

## [Decision Letter · Decision Letter 1]

24 Oct 2019

PONE-D-19-17658R1

Genetic and morphological divergence in the warm-water planktonic foraminifera genus Globigerinoides

PLOS ONE

Dear Dr. Morard,

Thank you for submitting your manuscript to PLOS ONE. After careful consideration, we feel that it has merit but does not fully meet PLOS ONE’s publication criteria as it currently stands. Therefore, we invite you to submit a revised version of the manuscript that addresses the points raised during the review process.

I have now received the comments on the revised version of your Ms by an external reviewer that appreciated all the effort in improving the early version. The reviewer has however raised some points which would be considered before the final acceptance. Among them, you should carefully address the last comment on the chambers’ shape of G. ruber.

We would appreciate receiving your revised manuscript by Dec 08 2019 11:59PM. To enhance the reproducibility of your results, we recommend that if applicable you deposit your laboratory protocols in protocols.io, where a protocol can be assigned its own identifier (DOI) such that it can be cited independently in the future. For instructions see: http://journals.plos.org/plosone/s/submission-guidelines#loc-laboratory-protocols

We look forward to receiving your revised manuscript.

Kind regards,

Fabrizio Frontalini

Academic Editor

PLOS ONE

Reviewers' comments:

Reviewer's Responses to Questions

**Comments to the Author**

1. If the authors have adequately addressed your comments raised in a previous round of review and you feel that this manuscript is now acceptable for publication, you may indicate that here to bypass the “Comments to the Author” section, enter your conflict of interest statement in the “Confidential to Editor” section, and submit your "Accept" recommendation.

Reviewer #1: (No Response)

2. Is the manuscript technically sound, and do the data support the conclusions?

Reviewer #1: Partly

3. Has the statistical analysis been performed appropriately and rigorously? 

Reviewer #1: No

4. Have the authors made all data underlying the findings in their manuscript fully available?

Reviewer #1: Yes

5. Is the manuscript presented in an intelligible fashion and written in standard English?

Reviewer #1: Yes

6. Review Comments to the Author

Reviewer #1: I appreciate the detailed author’s reply to my first review. Whereas I agree to most of the author's points, I still have a couple of points to be addressed before publication of the manuscript:

Line 522, and following: Please check again, and change “Indo-Pacific” to “Indian and Pacific Oceans”

Line 587: carbon isotopes and trace elements in the calcite shell, better change to “calcareous” shell…

Lines 653-654: “The roundness of the chambers of G. ruber remains stable throughout its ontogeny (Fig 10).”—I still don’t agree to this statement and the following discussion on neoteny. Juvenile G. ruber have a rather compressed bean shape, which changes into a more spherical shape in the neanic an adult stage. While the data and images (poor quality in my copy) provided here may not show the change in shape, this change in shape (visible also in the development of volume) is still the case. The authors may ask their senior co-author G.-J. Brummer, who may know best.

7. PLOS authors have the option to publish the peer review history of their article (what does this mean?). If published, this will include your full peer review and any attached files.

Reviewer #1: No

---

## [Author Response · Author response to Decision Letter 1]

29 Oct 2019

In the following, the comments of the reviewers are indicated by these symbols ***…*** and our responses are indicated by these symbols >>> …. <<<. We specify when necessary the page number of the changes we have made in the modified version of the manuscript with track changes. 

***Reviewer #1: I appreciate the detailed author’s reply to my first review. Whereas I agree to most of the author's points, I still have a couple of points to be addressed before publication of the manuscript:

Line 522, and following: Please check again, and change “Indo-Pacific” to “Indian and Pacific Oceans”***

>>> We have made the changes following the reviewer’s request. Lines 522, 524, 533, 536, 538-539.<<<

Line 587: carbon isotopes and trace elements in the calcite shell, better change to “calcareous” shell…

>>> We have made the modification. Line 588.<<<

Lines 653-654: “The roundness of the chambers of G. ruber remains stable throughout its ontogeny (Fig 10).”—I still don’t agree to this statement and the following discussion on neoteny. Juvenile G. ruber have a rather compressed bean shape, which changes into a more spherical shape in the neanic an adult stage. While the data and images (poor quality in my copy) provided here may not show the change in shape, this change in shape (visible also in the development of volume) is still the case. The authors may ask their senior co-author G.-J. Brummer, who may know best.

>>> We are here describing the Raupian parameters that are descriptors of shape within the Globigerinoides genus. As it stands, such parameters are over-simplifying the complexity of a shape but have the merit to make the comparison between species easier, e.g. with numbers and not words. We agree that the evolution of the shape of G. ruber could be described following the reviewer’s words, but all the species have been subjected to the same analysis and we cannot help that G. ruber is the species displaying the least variation of the shape (Descriptor S) during its ontogeny. However, we agree with the reviewer that our formulation might not reflect other aspects of ontogenetic morphology and we have modified the statement in order to tone down this assertion. Lines 654-656.

Regarding the discussion on neoteny following the statement, we believe that we are entitled to express our ideas and interpretation of the patterns we observe even if the reviewer does not agree with us. The change in the pace of volume increase observed with G. ruber actually supports the idea that G. ruber follows a neotenic ontogenetic trajectory. Neoteny is defined as the retention of juvenile features (round chambers) in the adult without a reduction of the size. The increase of the size of the chamber mentioned by the reviewer may compensate for the fewer chambers in the analyzed specimen. As a result, this allowed the specimen to reach the typical adult size of G. ruber, similar to G. elongatus whilst G. tenellus which also has fewer chambers (but with similar chamber size to G. elongatus) is overall smaller.

We have had lengthy discussions on the question with all co-authors, among which G. -J Brummer and we all agreed on the content of the manuscript before submission, including the heterochrony and the relevance of using Raupian parameters to describe the development of members of the Globigerinoides genus. We consider that our manuscript is open to debatable interpretation and we are not here pretending that heterochrony is the definitive answer, only a potential explanation.<<<

---

## [Editor Report · Decision Letter 2]

1 Nov 2019

Genetic and morphological divergence in the warm-water planktonic foraminifera genus Globigerinoides

PONE-D-19-17658R2

Dear Dr. Morard,

We are pleased to inform you that your manuscript has been judged scientifically suitable for publication and will be formally accepted for publication once it complies with all outstanding technical requirements.

With kind regards,

Fabrizio Frontalini

Academic Editor

PLOS ONE
---

## [Editor Report · Acceptance letter]

14 Nov 2019

PONE-D-19-17658R2 

Genetic and morphological divergence in the warm-water planktonic foraminifera genus *Globigerinoides*

Dear Dr. Morard:

I am pleased to inform you that your manuscript has been deemed suitable for publication in PLOS ONE. Congratulations! Your manuscript is now with our production department. 

With kind regards,

on behalf of

Dr. Fabrizio Frontalini 

Academic Editor

PLOS ONE